behaviour/biophysics/biomathematics

wandering albatross, GPS tracking, dynamic soaring, wind shear, airspeed, flight trajectory

**Author for correspondence:**
Philip L. Richardson
e-mail: prichardson@whoi.edu

# Observations and models of across-wind flight speed of the wandering albatross

Philip L. Richardson[1] and Ewan D. Wakefield[2]

[1]Department of Physical Oceanography, MS#21, Woods Hole Oceanographic Institution, 360 Woods Hole Road, Woods Hole, MA 02543, USA
[2]Institute of Biodiversity Animal Health and Comparative Medicine, University of Glasgow, Graham Kerr Building, Glasgow G12 8QQ, UK

(iD) PLR, 0000-0001-9115-2445

Wandering albatrosses exploit wind shear by dynamic soaring (DS), enabling rapid, efficient, long-range flight. We compared the ability of a theoretical nonlinear DS model and a linear empirical model to explain the observed variation of mean across-wind airspeeds of GPS-tracked wandering albatrosses. Assuming a flight trajectory of linked, 137° turns, a DS cycle of 10 s and a cruise airspeed of 16 m s$^{-1}$, the theoretical model predicted that the minimum wind speed necessary to support DS is greater than 3 m s$^{-1}$. Despite this, tracked albatrosses were observed in flight at wind speeds as low as 2 m s$^{-1}$. We hypothesize at these very low wind speeds, wandering albatrosses fly by obtaining additional energy from updrafts over water waves. In fast winds (greater than 8 m s$^{-1}$), assuming the same 10 s cycle period and a turn angle (TA) of 90°, the DS model predicts mean across-wind airspeeds of up to around 50 m s$^{-1}$. In contrast, the maximum observed across-wind mean airspeed of our tracked albatrosses reached an asymptote at approximately 20 m s$^{-1}$. We hypothesize that this is due to birds actively limiting airspeed by making fine-scale adjustments to TAs and soaring heights in order to limit aerodynamic force on their wings.

## 1. Introduction

Albatrosses routinely fly long distances over the Southern Ocean, even around the world, while seldom flapping their wings and expending little energy compared to other birds [1–3]. They are thought to do so primarily by dynamic soaring (DS), a technique that uses the vertical gradient of wind velocity (wind shear) near the ocean surface to provide energy for sustained soaring [4–6]. It is important to quantify the relationship between flight speed and wind velocity because this is fundamental to understanding

**Figure 1.** Schematic of a wandering albatross flying in an across-wind direction using an S-shaped dynamic soaring manoeuvre consisting of a series of upwind and downwind turns through the boundary layer (redrawn after Sachs [6]). The bird extracts mechanical energy from the wind by climbing headed upwind and descending headed downwind. Wave heights are typically large in the Southern Ocean. Wind–wave interactions cause a more complicated instantaneous wind field than the average shown here, and waves themselves induce updrafts. Albatrosses appear to efficiently exploit these fine-scale variations in wind velocity, making modelling their flight challenging.

the movements, distribution and foraging ecology of albatrosses and predicting how these could respond to future changes in the global wind climate [7].

Although numerous trajectories of wandering albatrosses have been measured by GPS tracking, the relationship between mean velocity during sustained, directed flight (e.g. commutes between breeding colonies and foraging patches or during non-breeding migrations) and wind velocity remains poorly understood. This is because available low temporal resolution (less than $10^{-3}$ Hz) measurements of bird and wind velocity have a large variance and although albatrosses have been tracked at higher temporal resolutions (greater than or equal to 1 Hz) [8–10], these data have not yet been systematically analysed in this way. Previously, we analysed low-resolution GPS data to create polar diagrams of air and ground speeds as a function of wind velocity [11]. Here, we extend that work by investigating the across-wind flight speed of wandering albatrosses (*Diomedia exulans*) with numerical simulations using both a linear model and a nonlinear DS model.

Wandering albatrosses are highly adapted to soaring flight. Their wings span up to approximately 3.5 m (the largest of all birds) with a large aspect ratio (approx. 12–15), giving them the largest glide ratio of any bird of approximately 21 [12]. They lack sufficient musculature to sustain continuous flapping flight for long periods but have a shoulder lock that mechanically holds their wings outstretched so that little energy is expended while soaring [12]. In across-wind flight, a DS albatross follows an S-shaped sinusoidal trajectory consisting of alternating upwind and downwind turns. For example, a bird starts flying in an across-wind direction at low level, turns upwind and climbs across the wind-shear layer, turns downwind and descends back across the wind-shear layer, and then turns across-wind again (figure 1). Given sufficient wind, albatrosses can use this technique to soar in any direction relative to the wind including upwind by tacking like a sailboat, but across-wind flight is relatively fast and appears to be preferred [11,13,14].

High-resolution GPS trajectories of wandering albatrosses have provided some quantitative details of typical DS manoeuvres. For example, across-wind trajectories recorded at 10 Hz consisted of a series of serially executed approximately 60° turns [8] (figure 2). Average ground velocities were approximately 16 m s$^{-1}$ in average wind speeds of 8 and 17 m s$^{-1}$ (referenced to 10 m height). A larger series of flight trajectories was recorded at 1 Hz in winds of 8–15 m s$^{-1}$ [10]. The median turn angle (TA) of a long across-wind trajectory from this series was found to be 66° [15]. A histogram of the number of observations of different TAs [15] indicates a large variation of TAs ranging from zero up to approximately 145° (figure 3). The average across-wind velocity associated with these measurements is 20 m s$^{-1}$ averaged over 9 h [15].

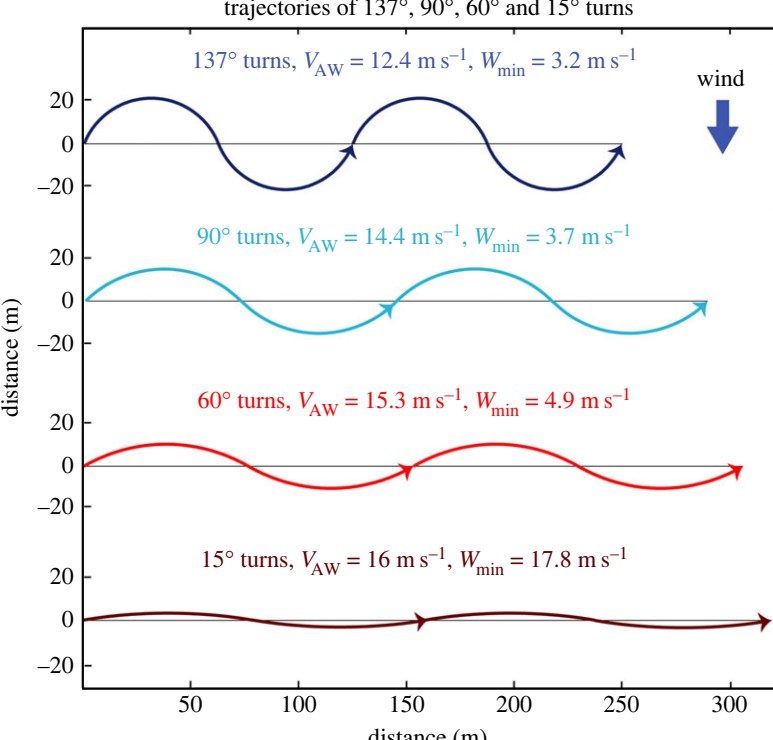

**Figure 2.** Plan view of idealized S-shaped across-wind trajectories through the air perpendicular to the wind velocity. Trajectories combine a series of 10 s dynamic soaring cycles of two circular turns alternating to the bird's right and left with an airspeed of 16 m s$^{-1}$. Turns located below the horizontal lines are assumed to be in the lower layer, where wind speed = 0; turns above the lines are in the upper layer, where wind speed = $W$. Trajectories cross the wind-shear layer (indicated by horizontal lines) with an angle equal to one-half of the turn angle (e.g. 45° for a 90° turn). $V_{AW}$ is the across-wind airspeed (magnitude of average across-wind velocity through the air), and $W_{min}$ is the minimum wind speed for dynamic soaring for the different turn angles. Note that across-wind airspeed $V_{AW}$ increases from 12 to 16 m s$^{-1}$ as turn angles decrease from 137° to 15° without any increase of along-trajectory airspeed $V$.

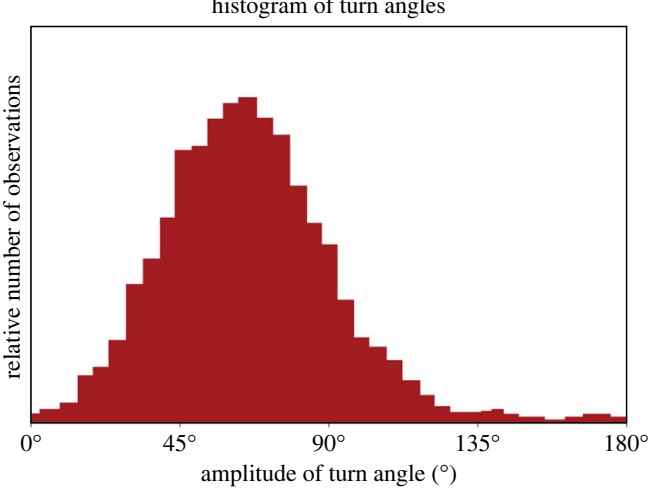

**Figure 3.** Amplitude of turn angles observed along the across-wind flight of a wandering albatross (redrawn from Bousquet *et al*. [15]). The trajectory was measured by Yonehara *et al*. [10]. The median turn angle is approximately 66° [15]; a large variability of turn angles is indicated by values ranging from zero up to approximately 145°. The across-wind flight velocity associated with this 650 km long trajectory is 20 m s$^{-1}$ averaged over 9 h [15].

Observations suggest that the typical period of a DS cycle (two consecutive turns in different directions) is 10–11.5 s [8,15,16]. The largest value 11.5 s, with a standard deviation of approximately 4 s [15], is associated with the TA data shown in figure 3. We are not aware of any studies that

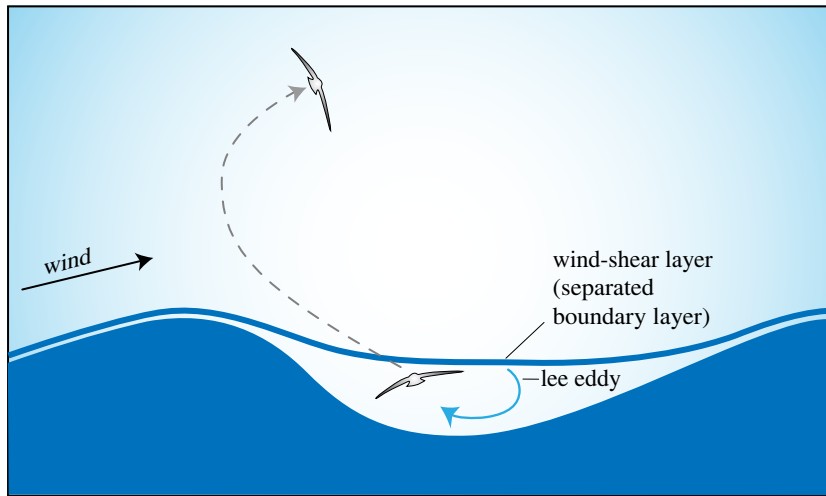

**Figure 4.** Schematic of an albatross 'gust soaring' (redrawn after Pennycuick [17]). Starting in a lee eddy (or separation bubble) located downwind of a sharp-crested wave, a bird climbs up through a thin wind-shear layer (separated boundary layer) that has detached from the wave crest. On crossing the wind-shear layer, the bird's airspeed abruptly increases, and the bird experiences a 'gust'. A lee eddy is a region of closed streamlines with clockwise circulation in this figure.

document correlations of TA and turn durations with wind speed or across-wind airspeeds, but based on these studies we assume that the typical cycle period in fast across-wind flight (approx. 20 m s$^{-1}$) is approximately 12 s.

Wind speed over the ocean generally increases with height above the average ocean surface. The largest gradient of the average wind velocity occurs within around 2 m of the average sea level [6]. Instantaneous wind profiles are more complex because of wind–wave interactions and turbulence. Low wind speeds occur in wave troughs and faster winds above wave crests. Fast wind blowing over the crest of a wave can separate from the crest forming a lee eddy in the wave trough. Large shear exists between wave trough and wave crest (figure 4). Wandering albatrosses appear to exploit this strong shear to gain energy for sustained soaring. Pennycuick [17] suggested that a bird's airspeed increases rapidly when climbing over a wave crest and encountering a gust of wind (figure 4). He referred to this as 'gust soaring'. The bird's airspeed also increases markedly during the descending, downwind phase of the cycle when re-entering the slow-moving layer of air in the wave trough.

The leading edge of a propagating ocean wave has an upward motion, which forces an updraft over the wave. We have often seen brown pelicans (*Pelecanus occidentalis*) in very low winds soaring long distances over the leading edge of swell waves propagating toward the coasts of California and Mexico. Occasionally these pelicans pull up over a swell wave, turn in an offshore direction, and descend over the leading edge of another swell wave thereby continuing to soar parallel to the coast. We have also seen northern fulmars (*Fulmarus glacialis*) soaring on the leading edges of swell waves in the North Sea. This flight manoeuvre is somewhat similar to the DS manoeuvre. Wandering albatrosses apparently use a similar technique to soar, possibly in combination with DS, when wind shear alone is not strong enough to entirely support the latter [12,13,18]. In addition, they can sustain flight by flapping their wings in low winds and small waves, but wing flapping is much more energetically expensive than soaring [2].

Visual observations of wandering albatrosses indicate they (i) use DS in sufficiently fast winds and small waves, (ii) use updrafts over sufficiently large waves in low wind and (iii) usually sit on the ocean surface when the wind is low and waves small [19,20]. The implication is that the birds can exploit both wind shear and updrafts over waves for sustained soaring, although separating the relative importance of the two sources of energy is difficult.

The purpose of our study is to test the adequacy of a simple theoretical model of DS, to predict the mean across-wind airspeed of wandering albatrosses and to revise hypotheses in the light of our findings. Our aims are, firstly, to investigate the relationship between the mean velocity of wandering albatrosses and wind velocity in order to gain a better understanding of their long-range flight characteristics; secondly, to use model simulations to interpret flight observations in terms of the period and TA of a DS cycle; and thirdly, to explain how and why wandering albatrosses fly so much slower than the fast speeds achieved by DS albatross-sized gliders.

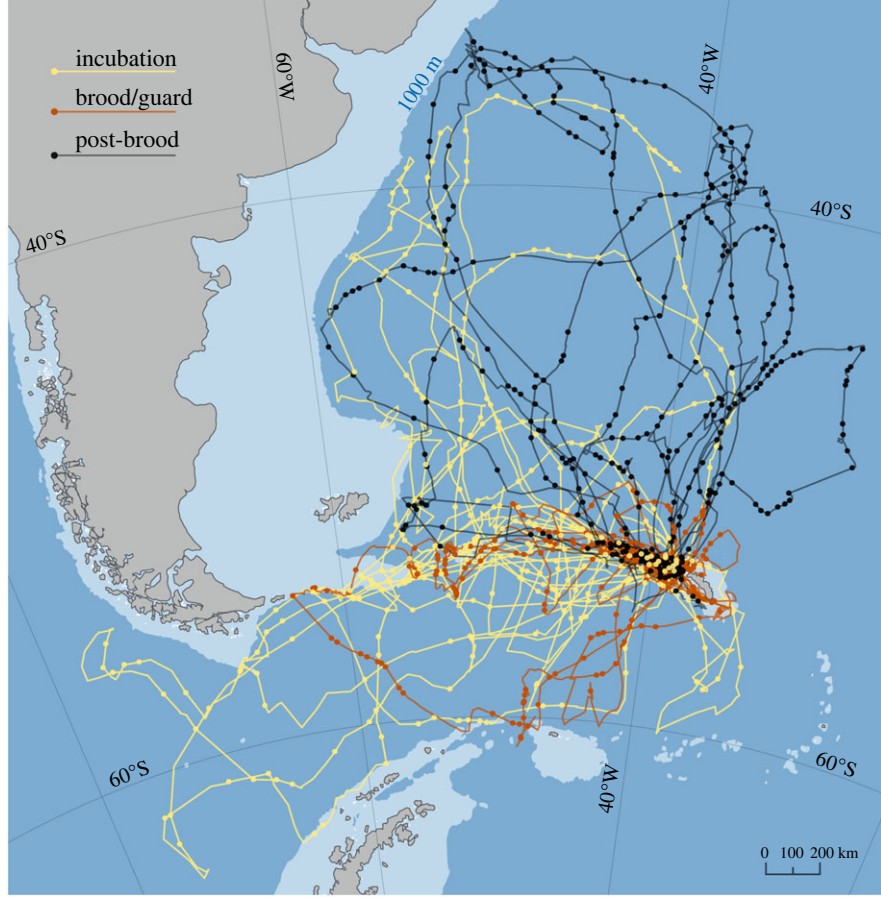

**Figure 5.** Trajectories of breeding wandering albatrosses GPS-tracked from Bird Island, South Georgia (54°S 38°W) during incubation (yellow, $n = 14$), brood/guard (red, $n = 15$) and post-brood (black, $n = 15$), where $n =$ the number of birds. Locations used in our analysis are highlighted with coloured dots and are those where track straightness was greater than or equal to 0.8 and the proportion of time on the water was less than 0.5 (trajectories of two birds that did not meet these criteria at any location are not shown). Six trajectories located north of South Georgia appear to loop counterclockwise centred roughly near 45°S 45°W.

## 2. Material and methods

### 2.1. Measurements of flight speed of the wandering albatross

Forty-six wandering albatrosses breeding on Bird Island, South Georgia were tracked by GPS during foraging trips made between February and September 2004 (figure 5). GPS positions were recorded at intervals of 0.5–2.0 h. In addition, activity loggers recorded saltwater immersion so that time actually spent flying could be estimated. We analysed only direct, sustained, bouts of flight and calculated ground velocity between GPS locations, resulting in 883 velocity measurements of 44 birds [14]. In the following, we quote mean values ± their standard errors. See appendix A for more information about measurements.

### 2.2. Measurements of wind velocity

We obtained wind data from the European Center for Medium-Range Weather Forecasts (ECMWF) on a grid of approximately 125 km in latitude by 75 km in longitude. Wind speed was estimated by interpolation to each bird location and reduced to a reference height of 5 m above mean sea level [14]. This is the median flight height for albatrosses observed from Bird Island [12], and was calculated assuming a logarithmic wind profile and a scale height of 0.03 [6].

### 2.3. Calculation of airspeed

The ground velocity of a bird is the vector sum of the velocity through the air (air velocity) and leeway velocity, which is defined as the bird's advection by the wind in the downwind direction. Airspeed, the

magnitude of air velocity, was calculated by subtracting leeway velocity from ground velocity measured by GPS [11]. The latter was measured at the scale of hours, whereas DS cycles occur at the scale of seconds. Hence, we use the term across-wind airspeed ($V_{AW}$) here and throughout to refer to the magnitude of across-wind air velocity averaged at the former scale. Leeway velocity was estimated using the slope parameter of a linear model of ground speed versus the component of wind velocity in the direction of flight $W\cos\theta$, where $W$ is the wind speed at a reference height of 5 m and $\theta$ is the angular difference between wind velocity and ground velocity. We calculated leeway velocity as being equal to the slope parameter (0.51) times wind velocity (see appendix A for details).

## 2.4. Theoretical dynamic soaring model of across-wind airspeed

In order to evaluate how well current theory predicts albatrosses flight speeds, we developed a DS model of across-wind flight in which the airspeed gained in climbing across the wind-shear layer is balanced by the loss of airspeed due to the drag of air acting on a bird, as described by Lissaman [4]. Two homogeneous layers were assumed in a wind-step model, a lower layer of zero wind speed and an upper layer of wind speed $W$. A thin wind-shear layer is sandwiched between the upper and lower layers. The zero wind speed in the lower layer represents regions of low wind speeds in wave troughs. The upper layer represents the fast, approximately free-stream wind blowing above wave crests.

The assumptions in the model are that: (i) a bird soars in a series upwind and downwind turns as part of an S-shaped trajectory oriented horizontally along a plane tilted upward into the wind at a small angle with respect to a level surface so that vertical motions can be ignored; (ii) the average airspeed and average glide ratio can be used to represent flight in the turns; and (iii) conservation of energy in each layer requires a balance between the sudden increase of airspeed caused by crossing the shear layer and the gradual loss of airspeed due to drag over each turn, resulting in energy-neutral soaring. The 10 s cycle period was assumed to be typical of that used across the range of observed airspeeds (see Introduction). As an example, we calculated wind speed as a function of airspeed using TAs of 90°. In order to investigate the effects of these parameters on theoretical flight performance in slow and fast winds, we also varied the assumed period and amplitude of the DS cycle over a realistic range. We note that a similar two-layer model for circular (hovering) flight agrees with results from more complicated aerodynamic models [21] and from radio-controlled model gliders DS in the lee of mountain ridges [22]. Further information about the model is given in appendix B.

# 3. Results

## 3.1. Empirical linear model of across-wind airspeed

A majority (67%) of the 883 observed flight directions in our dataset were located between 45 and 135° relative to the wind direction [11]. The average of all across-wind airspeeds (45–135°) is $12.4 \pm 0.1$ m s$^{-1}$, and the associated average wind speed is $9.2 \pm 0.1$ m s$^{-1}$. The average of the nine fastest across-wind airspeeds (fastest 1.5%) is $19.7 \pm 0.2$ m s$^{-1}$.

Across-wind airspeeds generally increased as a function of wind speed (figure 6). The empirical linear model of this relationship had an intercept of 8.8 ($\pm$ 0.3) ms$^{-1}$ and a slope of 0.39 ($\pm$ 0.03) (table 1), although there was considerable variation around this trend (coefficient of determination $R^2 = 0.2$). Airspeed increased by around 6 m s$^{-1}$ over the 15 m s$^{-1}$ increase of wind speed values (3–18 m s$^{-1}$). A quadratic model fitted to the data was virtually identical to the linear model and did not offer additional insight ($\Delta$AIC, simple versus quadratic model, 0.66).

The linear relationship between across-wind airspeeds and wind speed $W$ implies that airspeeds continue to increase up to at least $W = 20$ m s$^{-1}$, the upper limit of our observations. Extrapolating the linear model suggests that maximum airspeed could be reached when wind speed is approximately 29 m s$^{-1}$. However, our data indicate that in reality maximum across-wind airspeeds plateau at around 20 m s$^{-1}$ for $W > 8$ m s$^{-1}$, continuing at this level up to at least $W = 20$ m s$^{-1}$ (figure 6).

Residuals from the simple linear model were relatively large (RMSE = 2.5 m s$^{-1}$, table 1). We infer that this is partly due to variations in the speed of wandering albatrosses caused by differing wing loading, which is the weight of a bird divided by the area of its wings. For example, the mean across-wind airspeed of males was found to be $0.9 \pm 0.2$ m s$^{-1}$ faster than that of females [14], and the wing loading of males (148 N m$^{-2}$) was found to be 12.1% greater than that of females [23]. Cruise airspeed is proportional to the square root of wing loading [24], which suggests that the cruise airspeed of

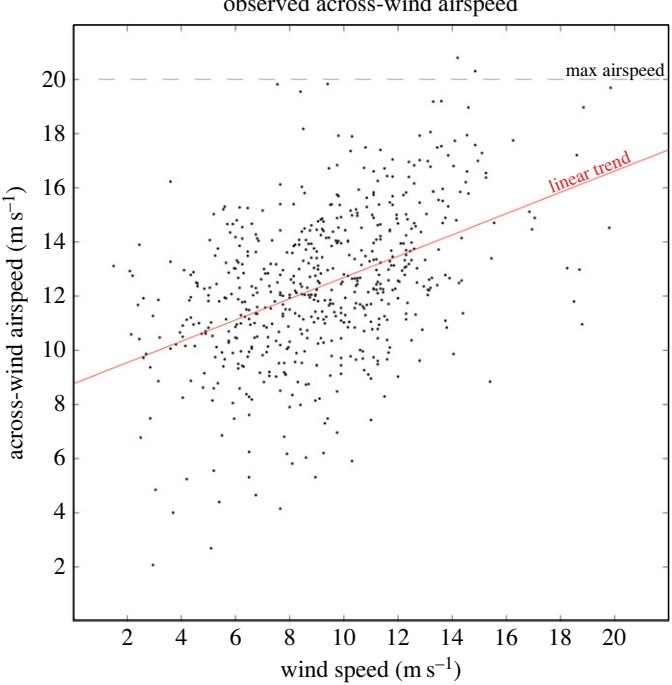

**Figure 6.** Mean across-wind airspeeds of GPS-tracked wandering albatrosses plotted as a function of wind speed. A linear model fitted to the data (slope = $0.39 \pm 0.03$) shows the average rate of increase of airspeed. The light dashed line indicates the approximate limit of maximum airspeeds near 20 m s$^{-1}$. Across-wind airspeeds are magnitudes of the bird's velocities through the air in directions between 45 and 135° relative to the downwind direction and averaged over 0.5–1.5 h.

**Table 1.** Performance of empirical models of wandering albatross across-wind airspeeds versus wind speed is given for different wind speed ($W$) ranges. Empirical models are simple linear models of across-wind airspeed versus wind speed. $W = 3.7$ m s$^{-1}$ is the minimum wind speed for a dynamic soaring trajectory with 90° turn angles and 10 s cycles. Root mean square errors (RMSE), coefficients of determination ($R^2$) and correlation coefficients ($r$) were calculated using observed across-wind airspeeds and wind speeds (see appendix A for details).

| type | model | $W$ (m s$^{-1}$) | $N$ | RMSE (m s$^{-1}$) | $R^2$ | $r$ |
|---|---|---|---|---|---|---|
| empirical linear model 1 | $V_{AW} = 8.76 \pm 0.31 +$ $(0.393 \pm 0.032)W$ | 2–20 | 596 | 2.5 | 0.20 | 0.44 |
| empirical linear model 2 | $V_{AW} = 8.34 \pm 0.34 +$ $(0.404 \pm 0.034)W$ | 3.7–20 | 573 | 2.4 | 0.20 | 0.45 |

males would also be approximately 0.7 m s$^{-1}$ greater than that of females. The wing loadings of different wandering albatrosses, which range from 117 to 170 N m$^{-2}$ and are caused by variations of body mass and food loading [23], correspond with a range of cruise airspeeds of approximately 3 m s$^{-1}$ for the birds. Some of the large residuals about the linear model could also be due to birds deviating from their general trajectories to forage between successive GPS positions. In addition, some residuals in speed could result from errors in wind speed caused by spatio-temporal mismatches in the tracking and wind data. Wind fields are inherently turbulent and evolve over similar time scales to our tracking data (hours), so that a significant part of the variance could be due to unresolved high-resolution variations in the wind field, which appear as airspeed variations in figure 6.

## 3.2. Dynamic soaring model predictions

Across-wind airspeeds $V_{AW}$ predicted by the DS model (equation (B 3)) are shown in figure 7 for a 10 s cycle period and 90° TAs. This curve is a compromise between the smaller observed approximately 60°

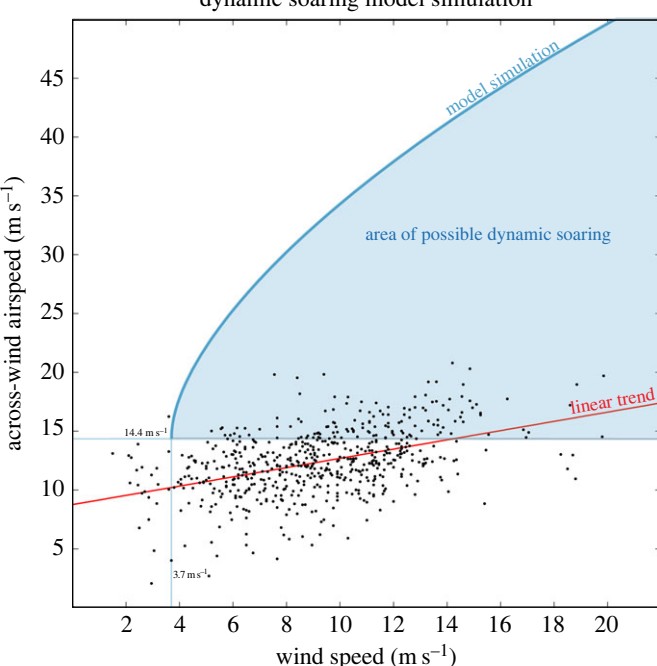

**Figure 7.** Observed across-wind airspeeds of wandering albatrosses plotted as a function of wind speed. The red line shows the empirical linear model 1 (as in figure 6). The blue line shows airspeed predicted by the DS model (equation (B 3)) for a wandering albatross dynamic soaring in an across-wind direction, with a cycle period of 10 s and 90° turns in a two-layer wind profile consisting of zero wind in the lower layer and a homogeneous wind $W$ in the upper layer. Model airspeeds are the speeds of predicted across-wind velocities ($V_{AW}$) through the air perpendicular to the wind direction. The shaded blue area represents the area of theoretically possible dynamic soaring. The lower limit of the blue area is the minimum across-wind airspeed of 14.4 m s$^{-1}$ associated with the minimum wind speed of 3.7 m s$^{-1}$ required for a 10 s 90° trajectory. Predicted curves for 8 and 12 s (not shown) are similar to the 10 s curve, and all three begin with an across-wind airspeed of 14.4 m s$^{-1}$. The 8 s curve is shifted slightly to the left of the 10 s curve to $W = 3.2$ m s$^{-1}$, and the 12 s curve is shifted slightly to the right of the 10 s curve to $W = 4.3$ m s$^{-1}$.

TAs in fast winds and the larger 112–137° TAs predicted by the DS model for minimum wind speeds $W_{min} \sim 3$–4 m s$^{-1}$.

### 3.2.1. Dynamic soaring in low wind speeds (less than 6 m s$^{-1}$)

Assuming a cycle period of 10 s, 137° turns, and a cruise speed of $V = 16$ m s$^{-1}$, the DS model predicts that a minimum wind speed $W_{min}$ of 3.2 m s$^{-1}$ would be required to sustain DS (figure 2). The model indicates that a bird could extend flight somewhat below 3.2 m s$^{-1}$ by decreasing the period of the DS cycle and simultaneously decreasing the amplitude of its TAs. For example, a bird soaring with an 8 s cycle period, 125° turns, and $V = 16$ m s$^{-1}$ could reduce $W_{min}$ to 2.9 m s$^{-1}$. This wind speed (2.9 m s$^{-1}$) is closer to but still above the minimum observed wind speeds. Therefore, a period of neither 10 s nor 8 s and associated TAs appear to explain the observations of very low winds less than 3 m s$^{-1}$. Two possible reasons for this discrepancy are explored in the Discussion section below.

### 3.2.2. Decrease of turn angles as wind increases

The DS model predicts little variation of $W_{min}$ as a function of TA centred near 137°. For example, values of $W_{min}$ vary by only 0.1 m s$^{-1}$ within a 50° range of TAs from 113 to 163°. Within this range the smallest TA (113°) is associated with the fastest across-wind airspeed ($V_{AW} = 13.5$ m s$^{-1}$) and also the smallest lift (approx. 1.2 $g$) associated with a steady banked turn. It seems possible that a bird could both increase across-wind airspeed and reduce lift by shifting to TAs below 137°.

In general, the DS model predicts that smallest values of $W_{min}$ are achieved with quite large TAs. For example, a $W_{min}$ of 4.2 m s$^{-1}$ could be achieved for a $V_{AW}$ of 20 m s$^{-1}$ using a TA of 112° and a cycle period of 10 s. In principle, a bird could use large TAs (112–137°) to maximize across-wind airspeed

and distance flown over the ocean for migration and foraging. However, the much smaller typical observed across-wind airspeeds (figure 7) and smaller TAs of approximately 60° (figure 3) reveal that in general the birds do not maximize their airspeeds with large TAs. Instead, the birds limit airspeed to be far below the large predicted values as shown in figure 7.

The curve of DS model $V_{AW}$ as a function of wind speed, predicted with a 10 s cycle and 60° turns (not shown), is similar to the 90° curve but is displaced towards higher wind speeds. The $W_{min}$ for 60° turns and $V = 16$ m s$^{-1}$ was found to equal 4.9 m s$^{-1}$. This indicates that for wind speeds greater than 4.9 m s$^{-1}$ maximum model across-wind airspeeds for 60° turns are significantly slower than those possible for 90° and larger TAs. This is mainly because a smaller increase of airspeed is gained from crossing the wind-shear layer with smaller angles relative to the across-wind direction (figure 2).

The DS model predictions and tracking data show that theoretically a bird could start DS with 137° turns in $W = 3.2$ m s$^{-1}$ and gradually shift to smaller TAs as $W$ increases. A minimum TA of 15° for DS was predicted with the DS model (equation (B 3)) for the point defined by maximum observed $W = 20$ m s$^{-1}$, maximum observed across-wind airspeed = 20 m s$^{-1}$ and a 10 s cycle period ( figure 7). The 15° TA corresponds to the minimum lift in a steady horizontal turn for $V_{AW} = 20$ m s$^{-1}$. We caution that in addition to adjusting TA, albatrosses could use other techniques to lower airspeed, such as adjusting the heights flown through the wind-shear layer. If other techniques were also being used, then the calculated minimum amplitude turn could be an underestimate of real TAs.

The shift to smaller TAs would be associated with an increase of $V_{AW}$ from 12.4 m s$^{-1}$ (137° turns) to 16.0 m s$^{-1}$ (15° turns) assuming a cycle period of 10 s (figure 2). Note that all this increase of $V_{AW}$ is due to the decrease of TA since $V = 16$ m s$^{-1}$ is constant. This increase in across-wind airspeed would seem to be advantageous for a bird foraging over the ocean because it could search a wider area in a given amount of time. Other advantages of the smaller observed turns compared to the 112–137° predicted turns are straighter trajectories (figure 2) and reduced acceleration and aerodynamic force, especially in fast flight. Therefore, shifting to the typical approximately 60° turns would tend to make DS less stressful and presumably easier for the birds.

### 3.2.3. Trajectory model

In order to verify that the predicted reduction of TAs as wind speed increases is consistent with the slope (0.40) of the empirical linear model of across-wind airspeed versus $W$ (figure 6), we developed a trajectory model (figure 2, equation (B 4)) that relates $V_{AW}$ to two parts—the average airspeed along a trajectory ($V$) and the magnitude of a TA. We assumed a linear model of the decrease of TAs from 137° to 15°. The trajectory model indicates that a 0.40 linear increase of across-wind airspeed could be achieved with an increase of $V$ from 16.0 to 19.2 m s$^{-1}$ over the wind speed range of 3.2–20 m s$^{-1}$ (figure 8). The two parts were found to contribute nearly equally in producing the total increase of across-wind airspeed of approximately 6.8 m s$^{-1}$ (figure 8). Therefore, we conclude that a linear decrease of TAs with increasing wind speed is plausible. Other variations of TA versus $W$ could also agree with the slope of the linear trend through airspeed data. This result is important because the relationship between airspeed and $W$ predicted by the DS model (figure 7) is curved and does not fit the data well. This is because the DS model is nonlinear and predicts the maximum possible airpeed for DS and because the birds limit their across-wind airspeeds to be much slower than predicted values.

### 3.2.4. Dynamic soaring in fast wind speeds ($W > 8$ m s$^{-1}$)

In wind speeds greater than 8 m s$^{-1}$, the DS model predicts much faster airspeeds than those observed, especially at the higher end of observed wind speeds (figure 7). Moreover, the apparent plateau of maximum observed airspeeds at a ceiling of approximately 20 m s$^{-1}$ (figure 6) means that neither the DS model nor the empirical linear model predicts airspeed well in fast winds. We explore the reasons for the large differences between the observed and predicted airspeeds in the Discussion section below.

### 3.2.5. Diagonal upwind dynamic soaring

For the same reasons as outlined above, albatrosses may also reduce their TAs as wind speed increases during upwind DS. The fastest upwind flight of a wandering albatross in typical winds tends to be along trajectories oriented along a diagonal line inclined by around 45° to the right or left of the wind direction [11,16,25]. We refer to this flight mode as diagonal upwind DS.

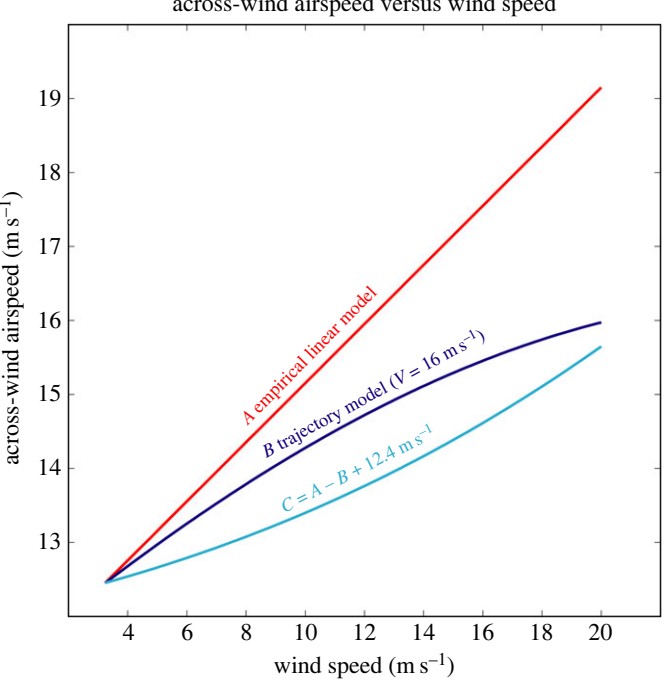

**Figure 8.** Illustration of how a linear increase of across-wind airspeed (as measured by GPS) could arise via the combination of two processes—simultaneous decrease in the amplitude of turn angle and an increase in average airspeed $V$ along a trajectory. Line A has the 0.40 slope of the empirical linear model fitted to the data (table 1) and starts at the point defined by $W_{min} = 3.2$ m s$^{-1}$ and across-wind airspeed $V_{AW} = 12.4$ m s$^{-1}$ associated with a trajectory with 137° turns. Curve B shows across-wind airspeed calculated with the trajectory model (equation (B 4)) assuming $V = 16$ m s$^{-1}$ and a linear decrease of turn angle from 137° at $W = 3.2$ m s$^{-1}$ to 15° at $W = 20$ m s$^{-1}$. Curve C is the difference between empirical linear model A and trajectory model B (plus 12.4 m s$^{-1}$). Curve C represents the part of the linear increase of across-wind airspeed due to an increase of airspeed $V$ from 16.0 to 19.2 m s$^{-1}$ over the same range of $W$.

A simple conceptual model of diagonal upwind DS based on visual observations [25] consists of a directly upwind climb across the wind-shear layer, a 90° turn to the right (say), an across-wind descent across the wind-shear layer, followed by a 90° turn to left toward an upwind direction. The trajectory would look similar to one shown in figure 2 for 90° turns but rotated 45° counterclockwise. A net increase of airspeed equal to $W$ would be obtained during two consecutive 90° turns as compared to an increase of airspeed equal to $1.4W$ for two 90° turns in the across-wind direction. Therefore, airspeed in a diagonal upwind direction would tend to be slower than airspeed in the across-wind direction. Moreover, diagonal upwind ground speed is reduced by opposing leeway.

Our DS model predicts that the minimum wind speed necessary for diagonal upwind flight is 5.2 m s$^{-1}$ (for 10 s cycles, with 90° turns and $V = 16$ m s$^{-1}$) and that maximum diagonal upwind airspeed is approximately 41 m s$^{-1}$ for a wind speed of 20 m s$^{-1}$. However, maximum observed diagonal upwind airspeeds are much slower, being approximately 17.5 m s$^{-1}$ [11]. Again, this discrepancy could be due to birds gradually reducing TAs below 90° as wind speed increases to avoid structural damage to their wings. The DS model predicts that the minimum possible TA for diagonal upwind DS at airspeed of 17.5 m s$^{-1}$ and wind speed of 20 m s$^{-1}$ is around 19°, which would also minimize lift in a steady banked turn. This hypothetical reduction of TA from 90° to 19° would reduce lift from around 1.8 $g$ to around 1.0 $g$ (for $W = 20$ m s$^{-1}$). To make progress directly upwind, a bird could tack like a sailboat, as wandering albatrosses have been observed to do when approaching putative foraging opportunities from directly downwind [26].

## 4. Discussion

Previously, we showed that maximum observed airspeeds (i.e. magnitude of air velocity) tend to occur in across-wind flight [11]. Indeed, the preponderance of observations in this direction suggests that albatrosses may prefer across-wind flight during sustained directed travel. This preference is probably

because a bird can efficiently extract energy from wind shear while maintaining a relatively fast average speed in this direction. Our analysis shows that the theoretical DS model overestimates observed albatross airspeeds in this direction in comparison to a simple empirical model. Below, we consider in more detail why this might be and how both theoretical and empirical insights gained from our analysis might be used to develop a more accurate model of albatross flight performance.

## 4.1. Minimum airspeed predicted by dynamic soaring model

A comparison of the predicted DS model across-wind airspeeds $V_{AW}$ and observed across-wind airspeeds shows that in low winds $V_{AW}$ differs from observed airspeed in two notable ways. First, the DS model predicts that minimum wind speeds of $W_{min} = 2.9$–$3.2$ m s$^{-1}$ are required for DS cycles with turns of 125–137° and periods of 8–10 s. These values suggest that DS cannot be sustained below a $W_{min}$ of around 3 m s$^{-1}$. However, numerous observed airspeeds are slower than 3 m s$^{-1}$. Second, $V_{AW}$ decreases rapidly as wind decreases from around 6 m s$^{-1}$ down to $W_{min}$ of 3.7 m s$^{-1}$ as shown in figure 7 for 90° turns (for example) unlike the trend of observed airspeeds. We interpret these differences to be due in part to birds exploiting updrafts over waves to supplement airspeed gained by DS in low winds. The combination of the two sources of energy could cause a more gradual decrease of observed airspeeds as $W$ decreases and enable soaring down to $W \sim 2$ m s$^{-1}$. However, we caution that the low observed airspeeds could also have been caused by errors in the wind data including spatio-temporal mismatches between the GPS and wind data.

A common perception is that an updraft over a wave is caused mainly by wind blowing up the windward face of a wave [12,21,27]. However, the causes and structures of updrafts are considerably more complicated than this and include air displaced upwards by the orbital velocity of a wave surface and vertical velocities from wind–wave interactions. These can occur simultaneously, their effects adding to and subtracting from each other in complicated ways. For example, a fast wind blowing over a relatively slow wave can cause an updraft over the windward wave face, but the updraft is countered somewhat by a downward orbital velocity. Leeward of the wave crest and centred just upwind of the wave trough a lee eddy can form causing an updraft, which augments an upward orbital velocity located there (figure 4). A bird flying horizontally in an updraft over waves could gain altitude (potential energy) from the wind, which could be used to balance the loss of energy due to drag. A more complete discussion of soaring in updrafts and relevant references are given in [21].

The gain of energy associated with updrafts over waves was estimated crudely by assuming a bird spends around half of each manoeuvre in a 1 m s$^{-1}$ updraft, which would result in an average vertical velocity of 0.5 m s$^{-1}$ [21]. This velocity would result in a height gain of approximately 2.5 m assuming horizontal flight through the air. Similarly, a bird could gain a height of 9 m from DS through an increase of wind speed of 5 m s$^{-1}$ across the wind-shear layer or a height gain of 20 m for a wind speed increase of 10 m s$^{-1}$. These values, which are estimated for typical conditions of the Southern Ocean, suggest that DS could provide around 80–90% of the total energy gain by both methods combined. Since in these typical conditions an albatross could gain an excess of energy using DS compared to what is needed for sustained soaring, there would appear to be little incentive for a bird to exploit updrafts over waves.

## 4.2. Fast airspeeds predicted by dynamic soaring model

The DS model overestimated observed across-wind airspeeds by a large amount—approximately 30 m s$^{-1}$ in fast winds ($W > 15$ m s$^{-1}$, figure 7). This overestimation is caused partly because the model predicts the maximum across-wind airspeed possible given the assumed DS cycle with 10 s period and 90° turns. Birds could easily fly slower by reducing the energy gained from wind shear. For example, a bird could soar across-wind at a constant 14.4 m s$^{-1}$ in increasing wind speed by limiting across-wind airspeed gained from wind shear (figure 7). The large predicted airspeeds could also be due to unrealistic assumptions, which perhaps overestimate airspeed gains and underestimate drag. It is also probable that model airspeeds, which represent averages over 10 s, tend to overestimate the much longer (0.5–1.5 h) averages of observed airspeeds, especially if the birds occasionally deviated from a straight track in to forage.

Because the predictions of our DS model's fast airspeeds disagree with the observations of wandering albatrosses (figure 7), one could infer that the model is incorrect. However, our DS model correctly predicts the extremely fast observed airspeeds, reaching 245 m s$^{-1}$, of radio-controlled gliders DS near mountain ridges [22] (www.rcspeeds.com). The combination of fast speeds and small periods of approximately 2–3 s of the circular trajectories required to reach these speeds results in extremely

large accelerations, greater than 100 times gravity, which require very strong wings. Clearly, wandering albatrosses limit their airspeed well below what is possible in DS, suggesting that our DS model might be more applicable to very strong albatross-like gliders soaring over the ocean. The implication is that in principle wandering albatrosses could fly much faster than the observations indicate but that the birds limit their airspeeds.

We hypothesize that wandering albatrosses limit their maximum across-wind airspeeds to approximately 20 m s$^{-1}$ in higher wind speeds (and greater wind turbulence), probably to keep the aerodynamic force on their wings during DS well below the mechanically tolerable limits of wing strength. Given the complex field of wind waves and swell waves often present in the Southern Ocean, it is also possible that birds find it increasingly difficult to coordinate DS manoeuvres at faster speeds. In order to limit maximum airspeed to 20 m s$^{-1}$ in fast winds, birds probably alter the DS manoeuvre in order to gain just enough airspeed to balance drag at an airspeed of approximately 20 m s$^{-1}$. Birds could probably do this by adjusting the height range traversed through the local instantaneous wind profile generated by wind–wave interactions and by adjusting the amplitude of their TAs (see appendix B).

Because of the large lift-to-drag ratio (approx. 21) of wandering albatrosses the largest component of aerodynamic force is lift. Lift of a bird in a steady banked turn increases with both the airspeed and amplitude of the TA and decreases with the period of the DS cycle (equation (B 5)). An extreme example is that in a wind speed of 20 m s$^{-1}$ the DS model predicts a maximum (average) across-wind airspeed of approximately 50 m s$^{-1}$ for a 10 s cycle with 90° turns (figure 7), which results in a lift of around 2.0 $g$ and bank angle of around 60°. If a bird limited across-wind airspeed to the maximum observed 20 m s$^{-1}$ and typical observed 60° turns the lift would be around 1.1 $g$ and bank angle around 24°. In this example, the large reduction of lift was caused by the combination of reduced airspeed and reduced TA. If the cycle period were to increase to 12 s near the 20 m s$^{-1}$ maximum across-wind airspeed then the lift would decrease slightly more by approximately 0.03 $g$, which is relatively small compared to the effects of airspeed and TA.

The relatively small lift for 60° TAs could explain why birds would prefer small TAs when flying in fast wind conditions, when turbulence can be large. A fast-flying bird encountering intense shear layers and horizontal and vertical gusts (and other types of turbulence) could be subjected to large abrupt increases in airspeed and angle of attack. These could cause sudden peaks of lift significantly larger than the average lift encountered in a typical turn, and their sum could be stressful for a bird.

For example, consider a hypothetical albatross DS with an average across-wind airspeed of 50 m s$^{-1}$ in wind speed of 20 m s$^{-1}$ and making 90° turns (figure 7). Assuming the lift was approximately 1.8 $g$ in a 57° banked turn before encountering the wind-shear layer, lift could suddenly jump to approximately 3.1 $g$ when the bird's airspeed increased on crossing the wind-shear layer, placing perhaps catastrophic stress on the bird's wings. Hypothetically, this force could be reduced if the bird quickly adjusted its wings and tail to reduce angle of attack. According to our model assumptions, this would coincide with the bird switching the direction of its bank angle from the left in an upwind turn to the right in a downwind turn, for example (figure 2). Presumably, however, this would be impractical because of the energetic cost of making large amplitude and quick changes in angle of attack and bank angle, especially considering that soaring is often sustained for long periods of time [28].

We have shown that the centripetal acceleration of a bird in a steady horizontal turn could be reduced by decreasing the amplitude of TAs. The acceleration could also be decreased by increasing the period of the DS cycle (equation (B 5)). If the assumption of constant period is relaxed, a bird could reduce acceleration by combining a decrease in TA with a simultaneous increase of cycle period.

GPS tracking (1 Hz) of a wandering albatross DS for a sustained period with a nearly constant 20 m s$^{-1}$ airspeed [15] indicated that cycle periods and TAs can be highly variable suggesting that albatross soaring is complex. The large variations of cycle period and TA are probably explained by the birds adjusting their DS manoeuvres to navigate over the complex wave patterns in the Southern Ocean, which consist of large local wind-driven waves plus swell waves that have propagated in from elsewhere. The birds also need to adjust their manoeuvres in the local wind field to gain sufficient energy for DS.

Clearly the correlations between TA, cycle period, airspeed and wind speed need to be established in order to understand what the birds actually do, especially in the region of slower winds and slower airspeeds where one might expect to find significant variations of TA and cycle period. Collecting the bird data needed to achieve this may already be achievable using a combination of high-resolution GPS, acceleration and video loggers [29,30] but contemporary wind and wave data at a similar resolution are likely to remain unobtainable for some time to come.

In summary, our observations and simulations lead us to hypothesize that albatrosses flying in relatively fast winds ($W > 8$ m s$^{-1}$) may limit the aerodynamic forces acting on their wings by limiting

maximum TAs to around 60°, by limiting DS cycles to a minimum period of approximately 12 s and by limiting airspeed by adjusting heights flown in the wind-shear layer. The result would be that maximum across-wind airspeed would remain less than 20 m s$^{-1}$, as we observed via tracking. The numerical experiments also indicate that a linear decrease of TAs with increasing wind speed from approximately 137° at $W = 3.2$ m s$^{-1}$ down to approximately 15° at $W = 20$ m s$^{-1}$ is consistent with the slope of observed across-wind airspeeds versus wind speeds (figure 6). A hypothetical increase of cycle period from 8 s up to 12 s could also be used to reduce lift in steady balanced turns. We view these hypotheses as one of the main outcomes of our analysis and trust this will stimulate others to test them as higher-resolution biologging and wind measurement techniques become available.

## 5. Conclusion

GPS measurements of the across-wind airspeed of wandering albatrosses were investigated in combination with satellite measurements of wind velocity in order to gain a better understanding of long-range flight speeds. A simple empirical linear model of across-wind airspeed as a function of wind speed predicts average across-wind airspeeds reasonably well over the 2–20 m s$^{-1}$ range of observed wind speeds (figure 6). The slope of the linear trend is in agreement with a predicted decrease of TAs from approximately 137° in low winds to approximately 15° in higher winds plus an increase of airspeed from 16.0 to 19.2 m s$^{-1}$.

A fairly simple nonlinear DS model was developed using the characteristics of observed DS manoeuvres (see appendix B). We found that this model considerably overestimates observed albatross airspeed, suggesting that airspeed is not limited by energy gain in DS, but instead is most likely limited by the mechanical strength of the bird's wings. Simulations using this model indicated that varying cycle period does not by itself generate variations in across-wind airspeed, so the linear trend of the model simulation of decreasing TAs would be similar with the inclusion of a hypothesized linear increase of cycle periods from 8 to 12 s.

We used model simulations to help interpret airspeed measurements. Specifically, the DS model predicts that for a 10 s cycle period and cruise airspeed of 16 m s$^{-1}$ the minimum wind for DS is approximately 3.2 m s$^{-1}$, that it occurs with a series of approximately 137° turns, and results in an across-wind airspeed of approximately 12.4 m s$^{-1}$. As wind speed and airspeed increase to large values the birds are inferred to reduce TAs to the typical observed approximately 60° in order to limit aerodynamic force on their wings. We conclude that in order to limit aerodynamic force, albatrosses could limit maximum across-wind airspeeds below around 20 m s$^{-1}$, limit maximum TAs to be less than 60° (in fast winds) and limit the minimum cycle period to around 12 s. Limiting lift in steady banked turns would be less stressful and easier for birds soaring in fast wind and encountering intense wind-shear layers and intense wind gusts, which could cause large spikes in lift.

Observations of across-wind airspeed extend down to a wind speed of approximately 2 m s$^{-1}$, which is significantly slower than the predicted minimum approximately 3 m s$^{-1}$ wind speed for DS. We conclude that the birds exploit both wind shear and updrafts over waves in order to soar in low wind speeds of approximately 2–3 m s$^{-1}$.

Ethics. We did not gather field data or track albatrosses as part of this paper. Instead, we used data obtained during 2004 by the staff of the British Antarctic Survey. These are archived in the BirdLife International Seabird Tracking Database. GPS tracking was ethically approved by the Government of South Georgia and the South Sandwich Islands and by the British Antarctic Survey (permit BAS 03-04).

Data accessibility. As part of this paper we used wandering albatross tracking data from 2004 with permission from Richard Phillips; the data are archived in the BirdLife International Seabird Tracking Database and were downloaded from http://seabirdtracking.org/mapper/?dataset_id=460. The basic wind data analysed were downloaded from the European Centre for Medium-Range Weather Forecasts archive, via http://apps.ecmwf.int/datasets/data/interim-full-daily/levtype=sfc/. The processed data supporting this article including GPS locations, wandering albatross velocities and associated wind velocities were compiled by E.D.W. and are available from the Dryad Digital Repository: https://doi.org/10.5061/dryad.zs7h44j96 [31].

Authors' contributions. P.L.R.: conceptualization, formal analysis, investigation, methodology, writing—original draft, writing—review and editing; E.D.W.: data curation, formal analysis, investigation, methodology, software, validation, writing—original draft, writing—review and editing.

All authors gave final approval for publication and agreed to be held accountable for the work performed therein.

Conflict of interest declaration. We declare we have no competing interests.

Funding. Funding was provided by the Woods Hole Oceanographic Institution emeritus fund and the UK Natural Environment Research Council (grant NE/M017990/1).

**Acknowledgements.** We thank Richard Phillips for providing the tracking data and fieldworkers, particularly Isaac Forster and Ben Phalan, who assisted with instrument deployments at Bird Island. We also thank the ECMWF for providing wind data and Gareth Marshall for facilitating its provision. Natalie Renier and Tim Silva helped create the figures. Gabriel Bousquet generously gave us and commented on a series of turn duration, TA and ground speed of wandering albatrosses that he derived from 1 Hz GPS positions measured by Yonehara *et al.* [10].

# Appendix A. Details of the measurements of wind speed and across-wind airspeed of the wandering albatross

## A.1. Satellite wind velocity

We obtained wind data, consisting of 6-hourly zonal and meridional wind speed components at a nominal height of 10 m above sea level, from the ECMWF on a reduced Gaussian grid (minimum node spacing at Bird Island, South Georgia (54°S), 125 km in latitude × 75 km in longitude). These data were produced by assimilation and reanalysis of observations of the global atmosphere, including wind speeds measured using the SeaWinds scatterometer aboard the QuikSCAT satellite, and observations from marine and terrestrial platforms [32]. The data are published as ERA-Interim dataset, available for download at http://apps.ecmwf.int/datasets/data/interim-full-daily/levtype=sfc/. We identified wind speed estimates nearest in time to each bird location and estimated wind speeds at these locations by linear interpolation between the two most spatially proximate grid points. We then reduced wind speeds to a reference height of 5 m above mean sea level, the median flight height for albatrosses observed from Bird Island [12], assuming a logarithmic average wind profile and a scale height of 0.03 [6].

## A.2. GPS velocity

The albatross tracking data, and the extraction and manipulation of data on wind speed and direction are described in detail by Wakefield *et al.* [14] and Richardson *et al.* [11]. In brief, 24 male and 22 female wandering albatrosses breeding on Bird Island, South Georgia were tracked by GPS during foraging trips made between February and September 2004. Birds were caught at the nest and BGDL-II GPS loggers (mass 67 g, dimensions $42 \times 71 \times 31$ mm$^3$) [33] were attached to their mantle feathers. In addition, an activity logger recording saltwater immersion was attached to a plastic ring placed around the tarsus. Total instrument mass, including attachment materials, was 0.6% of mean body mass, well below the 3% limit recommended for biologging studies on seabirds [34]. GPS loggers recorded locations to an accuracy of less than or equal to 10 m, at a temporal resolution of 2 h during incubation, 30 min during brood-guard and 60 min during post-brood chick-rearing. Activity loggers tested for saltwater immersion every 3 s and recorded a value between 0 and 200, representing the proportion of time wet, in 10-min blocks [34]. Devices were deployed for single foraging trips, after which birds were recaptured and the loggers removed. Logger deployment did not cause any observed injury, distress or adverse changes in behaviour. In this paper, we used wandering albatross tracking data from 2004, which are archived in the BirdLife International Seabird Tracking Database and can be downloaded from http://seabirdtracking.org/mapper/?dataset_id=460.

We analysed only direct, sustained, bouts of flight, which are defined as those during which straightness was greater than or equal to 0.8 and the proportion of time on the water was less than 0.5. Following Wakefield *et al.* [14], straightness for the *i*th location $L$ is the straight-line distance between locations $L_{i-1}$ and $L_{i+2}$ divided by the along track distance between these locations. Ground velocity between GPS locations was calculated and corrected for time actually spent in flight. A total of 883 velocity measurements was obtained of which a subset of 593 (67%) was used in the present analysis of magnitudes of across-wind airspeeds (directions of 45–135° relative to the wind direction).

The processed data supporting this article (GPS locations, wandering albatross velocity, associated wind velocity, etc.) were assembled by E.D.W., were submitted to the Dryad Digital Repository, and can be downloaded from https://doi.org/10.5061/dryad.zs7h44j96 [31].

## A.3. Leeway model and calculation of across-wind airspeed

We calculated the relative direction of albatross ground velocity with respect to wind velocity. We assumed symmetry in ground speeds in the across-wind direction (i.e. mean speed is the same, regardless of whether the birds are flying to the right or left of the wind direction) and combined all

observations into directions from zero (downwind) to 180° (upwind) relative to the wind direction (see Richardson *et al.* [11] for more details).

The ground velocity of a bird is the vector sum of the average velocity through the air (air velocity) and its leeway velocity, which is defined as the bird's advection by the wind in the downwind direction. Leeway tends to increase downwind ground velocity and decrease upwind ground velocity. In order to investigate how a bird's ground velocity varies with wind velocity it is helpful to analyse how both the bird's air velocity and leeway velocity vary as a function of wind velocity. In the case of DS birds, leeway velocity is usually different from the wind velocity at typical assumed reference heights because of the vertical shear of wind velocity near the ocean surface. Rather, leeway velocity is equal to the average wind velocity encountered by a bird as it soars vertically through the boundary layer.

In order to estimate leeway velocity, we assumed that it is directly proportional to wind velocity and calculated leeway velocity from the observed ground velocities and corresponding wind velocities. This same assumption was used as a basis to calculate fine-scale estimates of wind velocity from 1 Hz GPS measurements of the DS manoeuvres of seabirds [10]. Specifically, we modelled ground speed (magnitude of ground velocity) as a linear function of the component of wind velocity in the direction of ground velocity: ground speed (m s$^{-1}$) = $\alpha + \beta W \cos\theta$, where $W$ is the wind speed at a reference height of 5 m and $\theta$ is the relative angle between wind velocity and ground velocity [14].

Intercept $\alpha$ estimates the mean ground speed and the slope parameter $\beta$ is the fraction of the wind velocity causing leeway. $\beta$ was found to equal 0.51 (±0.02). This formulation implicitly assumes that ground speed does not vary with wind speed or the relative direction of the wind except through the second term, which represents leeway.

We calculated leeway velocities as being equal to the slope parameter (0.51) times wind velocities. Air velocities were calculated by subtracting leeway velocities from GPS-derived ground velocities. We also calculated the relative direction for each air velocity (i.e. the orientation with respect to the downwind direction). Note that ground velocity can be modelled by adding leeway velocity to air velocity after the variation of air velocity as a function of wind speed has been determined.

The slope parameter indicates that the effective leeway velocity is approximately one-half of the wind velocity at 5 m height. This effective leeway is lower than the wind velocity at 5 m height probably because albatrosses soar in the wind-shear boundary layer and spend considerable amounts of time below a height of 5 m and in wave troughs shielded from the full force of the wind.

# Appendix B. Dynamic soaring model

A bird climbing headed upwind across the wind-shear layer was assumed to increase airspeed by $\Delta V$, the component of wind velocity opposite to the direction of flight. $\Delta V$ was assumed to equal $W\sin(\text{TA}/2)$, where $W$ is the wind speed in the upper layer, TA is the amplitude of turn angle in degrees and TA/2 represents the angle with which the bird crosses the shear layer relative to the across-wind direction (figure 2). A similar increase of airspeed would be gained by a bird descending downwind across the wind-shear layer.

The sink rate $V_z$ of a bird at constant airspeed $V$ was used to model the decrease of airspeed under the assumption of constant height. To do this, the rate of change of kinetic energy, $\mathrm{d}/\mathrm{d}t(mV^2/2) = mV(\mathrm{d}V/\mathrm{d}t)$, at constant height was equated to the rate of change of potential energy, $\mathrm{d}/\mathrm{d}t(mgh) = -mgV_z$, at constant airspeed, where $V_z = -\mathrm{d}h/\mathrm{d}t$. The result indicates that $\mathrm{d}V/\mathrm{d}t = -g/(V/V_z)$, where $V/V_z$ represents values of the glide ratio (glide polar). Values of the glide ratio are closely equal to values of lift/drag ($L/D$) for $L/D$ values $\gg 1$ typical of albatross flight. Lift $L = C_L(\rho/2)V^2S$ and drag $D = C_D(\rho/2)V^2S$, where $C_L$ is the lift coefficient, $C_D$ is the drag coefficient and $S$ is the wing area.

$V/V_z$ is nearly constant in the relevant airspeed range $\Delta V$ centred at a particular average airspeed. Therefore, acceleration is nearly constant and airspeed decreases nearly linearly in time due to drag. For example, values of $V/V_z$ are within around 1% of the mean $V/V_z$ in the energy-neutral manoeuvre of a wandering albatross soaring with a cruise airspeed of 16 m s$^{-1}$. Thus, the total decrease of airspeed $\Delta V$ in a turn is given by $\Delta V = -0.5\,gt(V/V_z)$, where $t$ is the period of each DS cycle of two turns assumed to be 10 s [8,15,16].

Values of $V/V_z$ were modelled using the aerodynamic equations of motion for balanced circular flight [4,21] and a quadratic drag law, in which the drag coefficient is proportional to the lift

**Table 2.** Nomenclature for the dynamic soaring model.

| | |
|---|---|
| $C_L$ | lift coefficient |
| $C_D$ | drag coefficient |
| $D$ | drag |
| $g$ | gravity |
| $L$ | lift |
| $m$ | mass |
| $t$ | cycle period of two consecutive turns (s) |
| $T$ | period of 360° equivalent turn |
| TA | amplitude of turn angle (degrees) |
| $S$ | characteristic wing area |
| $V$ | model airspeed of wandering albatross along its flight trajectory |
| $V_{AW}$ | model across-wind airspeed (figure 2) |
| $V_c$ | cruise airspeed (16 m s$^{-1}$) at maximum glide ratio $(V/V_z)_{max} = 21.2$ |
| $V_z$ | sink rate due to drag |
| $\Delta V$ | increase of airspeed generated by crossing the wind-shear layer |
| $W$ | wind speed in upper layer of two-layer model |
| $W_{min}$ | minimum wind speed required for dynamic soaring |
| $\Delta W$ | increase of wind speed across the wind-shear layer |
| $\theta$ | angle between downwind direction and direction of flight when crossing the wind-shear layer |
| $\mu$ | bank angle |
| $\rho$ | density of air |

coefficient squared. In a steady banked turn the horizontal component of lift balances the centripetal acceleration and the vertical component of lift force balances gravity. Specifically, $V/V_z$ was modelled by

$$\frac{V}{V_z} = \frac{2(V/V_z)_{max}}{(V/V_c)^2 + (V_c/V \cos\mu)^2} , \tag{B1}$$

where $(V/V_z)_{max}$ is the maximum glide ratio at $V_c$ the associated cruise airspeed (airspeed of minimum drag) of an albatross in straight flight, $\mu$ is the bank angle (degrees) and cos$\mu$ is given by

$$\cos\mu = \sqrt{\frac{1}{(2\pi V/gT)^2 + 1}}. \tag{B2}$$

$T$ is the equivalent period of a 360° turn at a rate of $t$ seconds per cycle of two turns of angle TA° and is equal to $t(180°/\text{TA}°)$. The $2\pi V/gT$ term is the centripetal acceleration normalized by gravity. Combining equations (B 1) and (B 2) and assuming that the increase of airspeed $\Delta V$ in crossing the wind-shear layer is equal to $W\sin(\text{TA}°/2)$ we find that the DS model wind speed $W$, as a function of airspeed $V$, is given by the following equation:

$$W = \frac{gt}{4\sin(\text{TA}°/2)(V/V_z)_{max}} \left[ \left(\frac{V}{V_c}\right)^2 + \left(\frac{V_c}{V}\right)^2 + \left(\frac{2\pi V_c}{gT}\right)^2 \right]. \tag{B3}$$

Equation (B 3) indicates that for a particular bird with a given $(V/V_z)_{max}$ at $V_c$ in energy-neutral DS, the minimum wind speed required for soaring is a function of the average airspeed ($V$), the periods of $t$ and $T$ and the amplitude of the TA.

Equation (B 3) was solved to obtain minimum values of $W$ as a function of $V$ and also maximum values of $V$ as a function of $W$. The following characteristics of a wandering albatross in across-wind DS were used: cruise speed $V_c = 16$ m s$^{-1}$, which is the airspeed at the maximum glide ratio $(V/V_z)_{max} = 21.2$ in straight flight [35]; a DS cycle with a period of 10 s, consisting of two linked 90°

upwind and downwind turns; and an increase of airspeed of $W\sin(45°)$ in each crossing of the wind-shear layer.

Equation (B 3) indicates that the minimum wind speed $W_{min}$ in the upper layer that can support DS at $V = 16$ m s$^{-1}$ is 3.7 m s$^{-1}$. The model predicts that in a wind speed of 3.7 m s$^{-1}$ the bird's airspeed would be around 14.7 m s$^{-1}$ just before crossing the wind-shear layer, and would increase by 2.6 m s$^{-1}$ (3.7 m s$^{-1}$ times sin(45°) = 2.6 m s$^{-1}$) on crossing the wind-shear layer. The resulting 17.3 m s$^{-1}$ speed would then be reduced by drag to 14.7 m s$^{-1}$ just before the next crossing of the wind-shear layer.

The average airspeed $V$ in equation (B 3) is the scalar average of airspeed values experienced by the bird along a DS trajectory. In order to obtain model airspeed values that are equivalent to those derived from GPS measurements, we calculated the model across-wind airspeed ($V_{AW}$), which is the magnitude of the average model across-wind velocity through the air. $V_{AW}$ was calculated based on the shape of a trajectory by multiplying average airspeed $V$ by the distance along the straight line connecting the beginning and end of a circular arc of turn angle TA° and divided by the distance along the arc (figure 2). We refer to the resulting equation (B 4) as the trajectory model:

$$V_{AW} = V\left[\frac{360° \sin(TA°/2)}{\pi(TA°)}\right]. \quad (B 4)$$

The total aerodynamic force on a bird's wings is given by the sum of lift and drag. Since lift is significantly larger than drag for an albatross ($L/D \sim 21$), we use lift as a proxy of the aerodynamic force. Acceleration and lift (per unit mass) acting on the wings of a bird in a steady banked turn is equal to the vector sum of centripetal acceleration and gravity and is given by the load factor, which equals $1/\cos\mu$ (equation (B 2)). In the case of airspeed $V$ equal to 16 m s$^{-1}$ and 10 s cycles with 90° turns, the load factor is 1.12, the acceleration and lift (per unit mass) given in terms of gravity ($g$) is $1.12\,g$ and the average bank angle is 27°. Bank angle and lift increase as airspeed ($V$) increases and the period ($T$) of an equivalent 360° turn decreases (equation (B 2)).

The centripetal acceleration of a bird in a steady banked turn normalized by $g$ can also be expressed as

$$\text{Centripetal acceleration} = \left(\frac{\pi V}{90°g}\right)\left(\frac{TA°}{t}\right), \quad (B 5)$$

where $V$ is a bird's airspeed, TA is the turn angle, $t$ is the cycle period and $2TA/t$ is the turn rate. For a given $V$, acceleration could be reduced by reducing TA and/or by increasing $t$. We assumed that $t$ is approximately constant approximately 10 s as a function of wind speed [8,15,16] although this has not yet been documented with detailed analyses of high-resolution GPS and wind data. Assuming that $t = 10$ s, our DS model indicates the minimum wind speed for DS occurs with a TA = 137°. The median observed TA was found to be approximately 60° [15], which suggests that TA decreases with increasing wind speed and reduces acceleration. We modelled a linear decrease of TA as wind speed increases and compared results with the linear trend of $V$ versus $W$.

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
