## [Peer Review File · Royal Society Open Science]

Review History

Decision letter (RSOS-211364.R0)

Dear Dr Richardson

The Editors assigned to your paper RSOS-211364 "Observations and Models of Across-wind Flight Speed of the Wandering Albatross" have now received comments from reviewers and would like you to revise the paper in accordance with the comments from the Editors.

As you'll see, the Editors are of the view that the paper has technical merit, and should be suitable for further external review, but there is a concern that the scientific story that you are trying to tell is obscured by the current structuring of the paper. Specifically, the Editors feel that your work would be best served by the results and discussion being split into discrete segments of the manuscript, rather than being bundled into one section.

While the journal encourages a version of 'format-free initial submission' (<https://royalsociety.org/journals/authors/author-guidelines/#formatting>), we also ask that the paper be sufficiently that readers can easily navigate the story being told.

Given this, please can we ask that you make a further tweak to the structure to satisfy the Editors' concerns regarding the results and discussion?

Once you've made the modification, please resubmit and we'll ensure the paper is sent for external review.

on behalf of Dr Agustina Gómez-Laich (Associate Editor) and Miles Padgett (Subject Editor)
openscience@royalsociety.org

Associate Editor Comments to Author (Dr Agustina Gómez-Laich):

Associate Editor

Comments to the Author:

Dear authors,

This work presents some important and valuable information that adds to the body of literature on how albatrosses exploit wind shear by dynamic soaring. The obtained results suggest that albatrosses may use smaller turns angles than the theoretical optimum for fast flight in order to limit aerodynamic force on their wings.

I congratulate you for restructuring the manuscript, however in this new version the results and discussion are still integrated into one section when they should be into two separate ones. Considering this, I encourage you to submit the manuscript again. Briefly, RSOS asks a research article type to have the following main sections: Introduction, Materials and Methods, Results, Discussion, Conclusions. Attached you will be pleased to find a Word template for a research article type.

Kind regards,
Agustina

===PREPARING YOUR MANUSCRIPT===

Your revised paper should include the changes requested by the referees and Editors of your manuscript. You should provide two versions of this manuscript and both versions must be provided in an editable format:
one version identifying all the changes that have been made (for instance, in coloured highlight, in bold text, or tracked changes);
a 'clean' version of the new manuscript that incorporates the changes made, but does not highlight them. This version will be used for typesetting if your manuscript is accepted.

===PREPARING YOUR REVISION IN SCHOLARONE===

- If you are providing image files for potential cover images, please upload these at this step, and inform the editorial office you have done so. You must hold the copyright to any image provided.
- A copy of your point-by-point response to referees and Editors. This will expedite the preparation of your proof.

- Ensure that your data access statement meets the requirements at <https://royalsociety.org/journals/authors/author-guidelines/#data>. You should ensure that you cite the dataset in your reference list. If you have deposited data etc in the Dryad repository, please include both the 'For publication' link and 'For review' link at this stage.
- If you are requesting an article processing charge waiver, you must select the relevant waiver option (if requesting a discretionary waiver, the form should have been uploaded at Step 3 'File upload' above).
- If you have uploaded ESM files, please ensure you follow the guidance at <https://royalsociety.org/journals/authors/author-guidelines/#supplementary-material> to include a suitable title and informative caption. An example of appropriate titling and captioning may be found at https://figshare.com/articles/Table_S2_from_Is_there_a_trade-off_between_peak_performance_and_performance_breadth_across_temperatures_for_aerobic_scope_in_teleost_fishes_/3843624.

Author's Response to Decision Letter for (RSOS-211364.R0)

See Appendix A.

RSOS-211364.R1 (Revision)

Review form: Reviewer 1

Is the manuscript scientifically sound in its present form?

No

Are the interpretations and conclusions justified by the results?

No

Is the language acceptable?

Yes

Do you have any ethical concerns with this paper?

No

Have you any concerns about statistical analyses in this paper?

Yes

Recommendation?

Major revision is needed (please make suggestions in comments)

Comments to the Author(s)

This manuscript presents simulated flight trajectories of wandering albatrosses in relation to cross-winds, and the calculated speeds are compared with GPS-tracks of real albatrosses making flights around South Georgia. The main features of the loopy trajectories of dynamic soaring (DS) are turn angle and cycle period. The main question is how well DS can explain the ground speed in low and high wind speeds, and whether other mechanisms of extracting energy than DS come into play. Alternative mechanisms are gust soaring, swell soaring and slope soaring (the latter was introduced by Wilson (1975; Nature; a reference that strangely is missing in the present paper). There is a trend in the data that airspeed increases with wind speed, although there is an enormous scatter in the data (Fig 6). The main outcome of the paper (Fig 7) suggests that DS explains a minority of the observations. In the light of this it is somewhat surprising that the authors cling to DS as main explanation for albatross flight.

A main weak point of this analysis is that the various model results (varying wind speed, turn angles and cycle period) yield overall cross-country speed, which is compared with overall observed cross-country performance. Another weak point is that winds are not directly measured, but interpolated from weather data, which prevents the estimation of gust strength and the characteristics of the vertical wind gradient. The GPS data are not of sufficient time resolution (one position every 0.5-2 hours, which is good for overall flight routes but not sufficient to test the trajectories of DS cycles) that would be required to actually compare model assumptions with actual dynamic soaring trajectories. This may explain the large discrepancies between model calculations and data, which suggest that DS soaring is rarely the explanation of observed flight speeds. I think alternatives to DS are not discussed in the face of the results.

Specific comments

P3, L46 how was the L:D ratio of 21 obtained? Appears rather high unless in ground effect.

P14, L31 Wing loading is weight (mg) per unit wing area and therefore has unit N/m².

P16, L55 Unnecessary (tautologic) to say lift force, since lift is by definition a force. Same holds for drag.

P21, L8 How is across-wind different from the conventional cross-wind?

P21, L13 Don't you mean ground speed here?

P21, L40 small airspeed should be low airspeed.

P22, L18/19 245 m/s ??

P23, L28 How is it possible to reach airspeed of 50 m/s?

Review form: Reviewer 2

Is the manuscript scientifically sound in its present form?

No

Are the interpretations and conclusions justified by the results?

No

Is the language acceptable?

Yes

Do you have any ethical concerns with this paper?

No

Have you any concerns about statistical analyses in this paper?

Yes

Recommendation?

Reject

Comments to the Author(s)

Observations and Models of Across-wind Flight Speed of the Wandering Albatross

Overall, I found that the data presented was not sufficient to support the claims from the paper; I also found that a very strong and non-obvious hypothesis was made (namely keeping the dynamic soaring cycle time constant while modifying all other parameters), with a large influence on the conclusions of the article.

I recommend testing the hypotheses with data from [10].

The main stated paper contributions is a list of the following hypotheses:

- 1- that albatrosses utilize updrafts from water waves in low winds to sustain flight
- 2- that in high winds, albatrosses do not seek to maximize their airspeed, as that would result in excessive g-loading.
- 3- albatrosses limit their airspeed to 20 m/s
- 4- the way albatrosses limit their speed is by means of smaller turn angles.

1- and 2- have already been stated elsewhere e.g. in the beautiful [How do albatrosses fly around the world without flapping their wings? Richardson PL, 2011].

Claim 3- is based on a very few data points, so I don't find the statement very strong. In particular, I'd like to point the authors to [SUSTAINED FAST TRAVEL BY A GRAY-HEADED ALBATROSS (THALASSARCHE CHRYSOSTOMA) RIDING AN ANTARCTIC STORM, Catry P et al., 2004] where an albatross was recorded at an average groundspeed of 127 km/h (35 m/s). I recognize that ground speed and airspeed are not the same quantity, and that the albatross species are different in the two papers. Still, it makes me doubtful that wandering albatrosses actually limit their airspeed to 20 m/s.

The nonlinear DS model is based on the strong assumption that dynamic soaring cycles would be 10s long. However, there is no strong argument for keeping that value fixed. The only justification I found in the paper is "typical observed [half-period] of 5 s", and a reference to a "~11 s cycle in [15]". While it is true that [15] has mean cycles of 11-13s, the standard deviation on cycle duration is rather large.

In particular, the authors change the DS cycle turn amplitude while keeping period constant. I don't see why and to my knowledge it is not supported by data.

The authors make the hypothesis that with increasing airspeed the turn angle is reduced (with the implicit assumption that cycle time be kept constant). However, the data they use does not provide clear evidence of it. This said, I believe that the open data from [10] would allow actually testing the authors' hypothesis (hypothesis: cycle period is independent of turn amplitude -- I personally believe that the hypothesis does not hold and would instead bet that there is a negative correlation between turn amplitude and cycle time.).

This is an important point because fixing the cycle time to a constant value has a tremendous influence on all the conclusions of the paper. In particular, if one chose $t \sim TA$ in equation (B3), one would see that W is smaller for smaller values of TA .

In figure 6 (the main figure of the article), there is a log of scatter. I'm curious whether the scatter is in part due to the breadth of flight directions "45 to 135". It would be nice to color-code the scatter point with flight direction. Is there a pattern?

Decision letter (RSOS-211364.R1)

Dear Dr Richardson

The Editors assigned to your paper RSOS-211364.R1 "Observations and Models of Across-wind Flight Speed of the Wandering Albatross" have now received comments from reviewers and would like you to revise the paper in accordance with the reviewer comments and any comments from the Editors. Please note this decision does not guarantee eventual acceptance.

Please submit your revised manuscript and required files (see below) no later than 21 days from today's (ie 21-Feb-2022) date. Note: the ScholarOne system will 'lock' if submission of the revision is attempted 21 or more days after the deadline. If you do not think you will be able to meet this deadline please contact the editorial office immediately.

Please note article processing charges apply to papers accepted for publication in Royal Society Open Science (<https://royalsocietypublishing.org/rsos/charges>). Charges will also apply to papers transferred to the journal from other Royal Society Publishing journals, as well as papers submitted as part of our collaboration with the Royal Society of Chemistry

(<https://royalsocietypublishing.org/rsos/chemistry>). Fee waivers are available but must be requested when you submit your revision (<https://royalsocietypublishing.org/rsos/waivers>).

on behalf of Dr Agustina Gómez-Laich (Associate Editor) and Miles Padgett (Subject Editor)
openscience@royalsociety.org

Associate Editor Comments to Author (Dr Agustina Gómez-Laich):

Comments to the Author:

Dear authors,

The manuscript entitled "Observations and models of across-wind flight speed of wandering albatross" has now been seen by two referees, both of whom considered that substantial revisions are necessary. Both reviewers have concerns about the statistical analyses. Reviewer #1 states several weak points, which are principally related to the fact that the resolution that the wind and the GPS data have are not sufficient to compare model assumptions with dynamic soaring trajectories. Additionally, in the face of the results this reviewer suggests discussing more in detail alternatives to dynamic soaring. Reviewer #2 states that the presented data is not sufficient to support the claims of the paper and suggests testing the hypothesis with open data from a previous study. This reviewer also questions the assumption that dynamic soaring cycles are 10 s long and is concerned with the fact that authors modify the dynamic soaring cycle turn amplitude but keep the period constant.

Reviewer comments to Author:

Reviewer: 1

Comments to the Author(s)

This manuscript presents simulated flight trajectories of wandering albatrosses in relation to cross-winds, and the calculated speeds are compared with GPS-tracks of real albatrosses making flights around South Georgia. The main features of the loopy trajectories of dynamic soaring (DS) are turn angle and cycle period. The main question is how well DS can explain the ground speed in low and high wind speeds, and whether other mechanisms of extracting energy than DS come into play. Alternative mechanisms are gust soaring, swell soaring and slope soaring (the latter was introduced by Wilson (1975; *Nature*; a reference that strangely is missing in the present paper). There is a trend in the data that airspeed increases with wind speed, although there is an enormous scatter in the data (Fig 6). The main outcome of the paper (Fig 7) suggests that DS explains a minority of the observations. In the light of this it is somewhat surprising that the authors cling to DS as main explanation for albatross flight.

A main weak point of this analysis is that the various model results (varying wind speed, turn angles and cycle period) yield overall cross-country speed, which is compared with overall observed cross-country performance. Another weak point is that winds are not directly measured, but interpolated from weather data, which prevents the estimation of gust strength and the characteristics of the vertical wind gradient. The GPS data are not of sufficient time resolution (one position every 0.5-2 hours, which is good for overall flight routes but not sufficient to test the trajectories of DS cycles) that would be required to actually compare model assumptions with actual dynamic soaring trajectories. This may explain the large discrepancies

between model calculations and data, which suggest that DS soaring is rarely the explanation of observed flight speeds. I think alternatives to DS are not discussed in the face of the results.

Specific comments

P3, L46 how was the L:D ratio of 21 obtained? Appears rather high unless in ground effect.

P14, L31 Wing loading is weight (mg) per unit wing area and therefore has unit N/m².

P16, L55 Unnecessary (tautologic) to say lift force, since lift is by definition a force. Same holds for drag.

P21, L8 How is across-wind different from the conventional cross-wind?

P21, L13 Don't you mean ground speed here?

P21, L40 small airspeed should be low airspeed.

P22, L18/19 245 m/s ??

P23, L28 How is it possible to reach airspeed of 50 m/s?

Reviewer: 2

Comments to the Author(s)

Observations and Models of Across-wind Flight Speed of the Wandering Albatross

Overall, I found that the data presented was not sufficient to support the claims from the paper; I also found that a very strong and non-obvious hypothesis was made (namely keeping the dynamic soaring cycle time constant while modifying all other parameters), with a large influence on the conclusions of the article.

I recommend testing the hypotheses with data from [10].

The main stated paper contributions is a list of the following hypotheses:

- 1- that albatrosses utilize updrafts from water waves in low winds to sustain flight
- 2- that in high winds, albatrosses do not seek to maximize their airspeed, as that would result in excessive g-loading.
- 3- albatrosses limit their airspeed to 20 m/s
- 4- the way albatrosses limit their speed is by means of smaller turn angles.

1- and 2- have already been stated elsewhere e.g. in the beautiful [How do albatrosses fly around the world without flapping their wings? Richardson PL, 2011].

Claim 3- is based on a very few data points, so I don't find the statement very strong. In particular, I'd like to point the authors to [SUSTAINED FAST TRAVEL BY A GRAY-HEADED ALBATROSS (THALASSARCHE CHRYSOSTOMA) RIDING AN ANTARCTIC STORM, Catry P et al., 2004] where an albatross was recorded at an average groundspeed of 127 km/h (35 m/s). I recognize that ground speed and airspeed are not the same quantity, and that the albatross species are different in the two papers. Still, it makes me doubtful that wandering albatrosses actually limit their airspeed to 20 m/s.

The nonlinear DS model is based on the strong assumption that dynamic soaring cycles would be 10s long. However, there is no strong argument for keeping that value fixed. The only justification I found in the paper is "typical observed [half-period] of 5 s", and a reference to a "~11 s cycle in [15]". While it is true that [15] has mean cycles of 11-13s, the standard deviation on cycle duration is rather large.

In particular, the authors change the DS cycle turn amplitude while keeping period constant. I don't see why and to my knowledge it is not supported by data.

The authors make the hypothesis that with increasing airspeed the turn angle is reduced (with the implicit assumption that cycle time be kept constant). However, the data they use does not provide clear evidence of it. This said, I believe that the open data from [10] would allow actually testing the authors' hypothesis (hypothesis: cycle period is independent of turn amplitude -- I personally believe that the hypothesis does not hold and would instead bet that there is a negative correlation between turn amplitude and cycle time.).

This is an important point because fixing the cycle time to a constant value has a tremendous influence on all the conclusions of the paper. In particular, if one chose $t \sim TA$ in equation (B3), one would see that W is smaller for smaller values of TA .

In figure 6 (the main figure of the article), there is a log of scatter. I'm curious whether the scatter is in part due to the breadth of flight directions "45 to 135". It would be nice to color-code the scatter point with flight direction. Is there a pattern?

===PREPARING YOUR MANUSCRIPT===

If you have been asked to revise the written English in your submission as a condition of publication, you must do so, and you are expected to provide evidence that you have received language editing support. The journal would prefer that you use a professional language editing service and provide a certificate of editing, but a signed letter from a colleague who is a fluent speaker of English is acceptable. Note the journal has arranged a number of discounts for authors using professional language editing services (<https://royalsociety.org/journals/authors/benefits/language-editing/>).

===PREPARING YOUR REVISION IN SCHOLARONE===

Author's Response to Decision Letter for (RSOS-211364.R1)

See Appendix B.

RSOS-211364.R2 (Revision)

Review form: Reviewer 1

Is the manuscript scientifically sound in its present form?

No

Are the interpretations and conclusions justified by the results?

No

Is the language acceptable?

Yes

Do you have any ethical concerns with this paper?

No

Have you any concerns about statistical analyses in this paper?

Yes

Recommendation?

Reject

Comments to the Author(s)

The authors have prepared a revision of the submitted manuscript. I have carefully read the authors replies to both my and the other reviewers comments, and it is clear from this that the authors have been unable to produce a publishable manuscript. The reply contains multiple "we believe.." explanations to many of the comments, which I do not consider sufficient. It is clear that the uncertainties regarding the simulations and suggested further analyses that the authors have not made for the revision, and that in combination with the mismatch between sampling rates and estimated quantities, suggest to me that publication should be deferred until a more satisfying analyses can be presented. The argument that appropriate wind data are not available at the necessary time and spatial resolution, does not implicate that one should go on as if the data available are useful anyway.

Review form: Reviewer 2

Is the manuscript scientifically sound in its present form?

Yes

Are the interpretations and conclusions justified by the results?

Yes

Is the language acceptable?

Yes

Do you have any ethical concerns with this paper?

No

Have you any concerns about statistical analyses in this paper?

No

Recommendation?

Accept with minor revision (please list in comments)

Comments to the Author(s)

Thank you for the thoughtful answer. I appreciate your further data analysis. In particular I appreciate that you more explicitly discuss that it is a hypothesis that demands more study.

While there are points where I still do not align with the authors, I am more comfortable if the article given the modifications, and author's responses message.

* There is still a spot in the article where the 10s-cycle hypothesis makes me uncomfortable with your analysis: the paragraph on "p25 of 86", line 14 ("In general, the DS model predicts that smallest values of..."). If you were not constraining the cycle duration to 10s, you would obtain a different conclusion (for instance, I bet that if you tried the exact same model with a smaller cycle type, for instance 8s, you'd get a smaller Win, wherein the optimal cycle would be completed at smaller turn angle.

* I also still have doubts on the paper, in the numerical value "20 m/s" for the upper airspeed of the albatross. I appreciate The author's reply regarding Catry and al., where the author ran a computation to prove the their "20 m/s" claim was compatible with the observations from Catry et al. Indeed the author does a pure algebraic addition, while the airspeed to consider should rather be $\sqrt{35^2 - 20.5^2} = 28$ m/s.

I fully agree with the author that "It is clear that the fastest observed ground speeds of albatrosses are an order of magnitude lower than those predicted by dynamic soaring theory"; I predict though that the "20 m/s" value is too low. I suspect that Yonehara's data could be analyzed further to show this -- but I understand if this is "left for future work".

Accordingly I would be delighted if it was made more explicit in the article that 20 m/s is not an absolute upper bound, but the maximum seen in that particular dataset (particularly in the abstract).

Review form: Reviewer 3

Is the manuscript scientifically sound in its present form?

No

Are the interpretations and conclusions justified by the results?

No

Is the language acceptable?

Yes

Do you have any ethical concerns with this paper?

No

Have you any concerns about statistical analyses in this paper?

No

Recommendation?

Major revision is needed (please make suggestions in comments)

Comments to the Author(s)

Thank you for the opportunity to review this very interesting and well written manuscript. The proposed model of across-wind flight speed of dynamic soaring albatrosses will be useful in shaping our understanding of this incredible flight mode.

Although I agree with most of the conclusions in this manuscript, I do not believe the concerns raised by the first two reviewers were adequately addressed. I do believe that albatrosses limit their airspeed to reduce the physical force on their wings, but I don't believe that the validation data (airspeed derived from coarse-scale GPS loggers) are adequate to test these predictions. The model predicts airspeed of a single DS cycles (i.e. seconds) whereas the validation data are on a much larger scale (minutes to hours). The analysis of 1 Hz data is presented in the rebuttal, but I could not see how this was incorporated in the manuscript. I believe that these data have to be presented to the reader to make clear that the GPS data we currently have is not necessarily adequate to test these hypotheses.

Secondly, the fixing of DS duration at 10 s was also raised by both reviewers. A possible suggestion would be to present models where $t = 4.5, 5, \text{ and } 5.5$, to illustrate the effect that varying cycle duration would have on the model predictions, e.g. Fig 7 with multiple model prediction lines.

Lastly, it might be worth expanding a bit on the effect that varying bank angles might have in context of this model. From what I could gather, the model assumes a constant banked turn. Recent studies have been able to measure fine-scale bank angle of free ranging dynamic soaring birds (see refs below). It would be interesting to read your thoughts on how future studies could look at bank angles and how these would influence the turn angles of a DS cycle at varying wind speeds.

References:

S. Schoombie, J. Schoombie, C. W. Brink, K. L. Stevens, C. W. Jones, M. M. Risi, P. G. Ryan, Automated extraction of bank angles from bird-borne video footage using open-source software. *J. Field Ornithol.* 90, 361–372 (2019).

Kempton, James A., et al. "Optimization of dynamic soaring in a flap-gliding seabird affects its large-scale distribution at sea." *Science Advances* 8.22 (2022): eabo0200.

Decision letter (RSOS-211364.R2)

Dear Dr Richardson

The Editors assigned to your paper RSOS-211364.R2 "Observations and Models of Across-wind Flight Speed of the Wandering Albatross" have now received comments from reviewers and would like you to revise the paper in accordance with the reviewer comments and any comments from the Editors. Please note this decision does not guarantee eventual acceptance.

Please submit your revised manuscript and required files (see below) no later than 21 days from today's (ie 06-Sep-2022) date. Note: the ScholarOne system will 'lock' if submission of the revision is attempted 21 or more days after the deadline. If you do not think you will be able to meet this deadline please contact the editorial office immediately.

on behalf of Dr Agustina Gómez-Laich (Associate Editor) and Miles Padgett (Subject Editor)
openscience@royalsociety.org

Associate Editor Comments to Author (Dr Agustina Gómez-Laich):

Associate Editor: 1

Comments to the Author:

Dear authors,

Thank you for this new version of the Ms entitled "Observations and models of across-wind flight speed of wandering albatross". I appreciate all the efforts you made answering the questions and concerns raised by the reviewers. The paper reads much more clearly now, however, the Ms it is not acceptable for publication in the present form.

The three reviewers raised concerns about the simulation and analysis, particularly regarding the dissimilarity of the spatial and time scale employed for the models and for the validation. Additionally, Reviewer#1 as well as Reviewer#3 raised concerns about the 10s-cycle hypothesis. Regarding this particular issue, Reviewer#3 suggests to present models where $t = 4.5, 5, \text{ and } 5.5$, to illustrate the effect that varying cycle duration would have on the model predictions. Reviewer#1 also has doubts about the numerical value "20 m/s" for the upper airspeed. Reviewer#3 also suggests expanding at least a bit on the effect that varying bank angles might have in context of the presented model.

Kind regards,
Agustina

Associate Editor: 2
Comments to the Author:
Dear authors,

I appreciate all the efforts you made answering the questions and concerns raised by the reviewers. The paper reads much more clearly now, however, since the corrections involved even preliminary analyses with a new data set, I suggested sending the Ms. back to review.

Reviewer comments to Author:
Reviewer: 2
Comments to the Author(s)

Thank you for the thoughtful answer. I appreciate your further data analysis. In particular I appreciate that you more explicitly discuss that it is a hypothesis that demands more study.

While there are points where I still do not align with the authors, I am more comfortable if the article given the modifications, and author's responses message.

* There is still a spot in the article where the 10s-cycle hypothesis makes me uncomfortable with your analysis: the paragraph on "p25 of 86", line 14 ("In general, the DS model predicts that smallest values of..."). If you were not constraining the cycle duration to 10s, you would obtain a different conclusion (for instance, I bet that if you tried the exact same model with a smaller cycle type, for instance 8s, you'd get a smaller Win, wherein the optimal cycle would be completed at smaller turn angle.

* I also still have doubts on the paper, in the numerical value "20 m/s" for the upper airspeed of the albatross. I appreciate The author's reply regarding Catry and al., where the author ran a computation to prove the their "20 m/s" claim was compatible with the observations from Catry et al. Indeed the author does a pure algebraic addition, while the airspeed to consider should rather be $\sqrt{35^2 - 20.5^2} = 28 \text{ m/s}$.

I fully agree with the author that "It is clear that the fastest observed ground speeds of albatrosses are an order of magnitude lower than those predicted by dynamic soaring theory"; I predict though that the "20 m/s" value is too low. I suspect that Yonehara's data could be analyzed further to show this -- but I understand if this is "left for future work".

Accordingly I would be delighted if it was made more explicit in the article that 20 m/s is not an absolute upper bound, but the maximum seen in that particular dataset (particularly in the abstract).

Reviewer: 1

Comments to the Author(s)

The authors have prepared a revision of the submitted manuscript. I have carefully read the authors replies to both my and the other reviewers comments, and it is clear from this that the authors have been unable to produce a publishable manuscript. The reply contains multiple “we believe..” explanations to many of the comments, which I do not consider sufficient. It is clear that the uncertainties regarding the simulations and suggested further analyses that the authors have not made for the revision, and that in combination with the mismatch between sampling rates and estimated quantities, suggest to me that publication should be deferred until a more satisfying analyses can be presented. The argument that appropriate wind data are not available at the necessary time and spatial resolution, does not implicate that one should go on as if the data available are useful anyway.

Reviewer: 3

Comments to the Author(s)

Thank you for the opportunity to review this very interesting and well written manuscript. The proposed model of across-wind flight speed of dynamic soaring albatrosses will be useful in shaping our understanding of this incredible flight mode.

Although I agree with most of the conclusions in this manuscript, I do not believe the concerns raised by the first two reviewers were adequately addressed. I do believe that albatrosses limit their airspeed to reduce the physical force on their wings, but I don't believe that the validation data (airspeed derived from coarse-scale GPS loggers) are adequate to test these predictions. The model predicts airspeed of a single DS cycles (i.e. seconds) whereas the validation data are on a much larger scale (minutes to hours). The analysis of 1 Hz data is presented in the rebuttal, but I could not see how this was incorporated in the manuscript. I believe that these data have to be presented to the reader to make clear that the GPS data we currently have is not necessarily adequate to test these hypotheses.

Secondly, the fixing of DS duration at 10 s was also raised by both reviewers. A possible suggestion would be to present models where $t = 4.5, 5, \text{ and } 5.5$, to illustrate the effect that varying cycle duration would have on the model predictions, e.g. Fig 7 with multiple model prediction lines.

Lastly, it might be worth expanding a bit on the effect that varying bank angles might have in context of this model. From what I could gather, the model assumes a constant banked turn. Recent studies have been able to measure fine-scale bank angle of free ranging dynamic soaring birds (see refs below). It would be interesting to read your thoughts on how future studies could look at bank angles and how these would influence the turn angles of a DS cycle at varying wind speeds.

References:

S. Schoombie, J. Schoombie, C. W. Brink, K. L. Stevens, C. W. Jones, M. M. Risi, P. G. Ryan, Automated extraction of bank angles from bird-borne video footage using open-source software. *J. Field Ornithol.* 90, 361–372 (2019).
 Kempton, James A., et al. "Optimization of dynamic soaring in a flap-gliding seabird affects its large-scale distribution at sea." *Science Advances* 8.22 (2022): eabo0200.

===PREPARING YOUR MANUSCRIPT===

If you have been asked to revise the written English in your submission as a condition of publication, you must do so, and you are expected to provide evidence that you have received language editing support. The journal would prefer that you use a professional language editing service and provide a certificate of editing, but a signed letter from a colleague who is a fluent speaker of English is acceptable. Note the journal has arranged a number of discounts for authors using professional language editing services (<https://royalsociety.org/journals/authors/benefits/language-editing/>).

===PREPARING YOUR REVISION IN SCHOLARONE===

- An individual file of each figure (EPS or print-quality PDF preferred [either format should be produced directly from original creation package], or original software format).
- An editable file of each table (.doc, .docx, .xls, .xlsx, or .csv).
- An editable file of all figure and table captions.

- Any electronic supplementary material (ESM).
- If you are requesting a discretionary waiver for the article processing charge, the waiver form must be included at this step.
- If you are providing image files for potential cover images, please upload these at this step, and inform the editorial office you have done so. You must hold the copyright to any image provided.
- A copy of your point-by-point response to referees and Editors. This will expedite the preparation of your proof.

- Ensure that your data access statement meets the requirements at <https://royalsociety.org/journals/authors/author-guidelines/#data>. You should ensure that you cite the dataset in your reference list. If you have deposited data etc in the Dryad repository, please include both the 'For publication' link and 'For review' link at this stage.
- If you are requesting an article processing charge waiver, you must select the relevant waiver option (if requesting a discretionary waiver, the form should have been uploaded at Step 3 'File upload' above).
- If you have uploaded ESM files, please ensure you follow the guidance at <https://royalsociety.org/journals/authors/author-guidelines/#supplementary-material> to include a suitable title and informative caption. An example of appropriate titling and captioning may be found at https://figshare.com/articles/Table_S2_from_Is_there_a_trade-off_between_peak_performance_and_performance_breadth_across_temperatures_for_aerobic_sc_ope_in_teleost_fishes_/3843624.

Author's Response to Decision Letter for (RSOS-211364.R2)

See Appendix C.

Decision letter (RSOS-211364.R3)

Dear Dr Richardson

On behalf of the Editors, we are pleased to inform you that your Manuscript RSOS-211364.R3 "Observations and Models of Across-wind Flight Speed of the Wandering Albatross" has been accepted for publication in Royal Society Open Science subject to minor revision in accordance with the referees' reports. Please find the referees' comments along with any feedback from the Editors below my signature.

Please submit your revised manuscript and required files (see below) no later than 7 days from today's (ie 25-Oct-2022) date. Note: the ScholarOne system will 'lock' if submission of the revision is attempted 7 or more days after the deadline. If you do not think you will be able to meet this deadline please contact the editorial office immediately.

on behalf of Dr Agustina Gómez-Laich (Associate Editor) and Miles Padgett (Subject Editor)
openscience@royalsociety.org

Associate Editor Comments to Author (Dr Agustina Gómez-Laich):
Associate Editor
Comments to the Author:
Dear authors,

Thank you for this new version of the Ms. I appreciate all the efforts you made answering the questions and concerns raised by the reviewers. I only have a few minor comments.

Introduction.

- 1) Even though this paper does not follow the classic structure of a manuscript, I suggest authors to refresh the readers at the end of the introduction which are the principal aims of the work.
- 2) Line 54-55, page 12 of 82 (tracked changes version). A space is missing between "to" and "16 m/s"
- 3) Line 54-55, page 13 of 82 (tracked changes version). "In" should be replaced by "is". That is to say "variability of turn angles is indicated instead of "in indicated".

Methods.

- 4) Line 41, page 17 of 82 (tracked changes version). Hence we use "the" term. "The" is missing.

Results.

- 5) Line 5, page 19 of 82. Please eliminate the ")" that is before [11]

- 6) Line 8-9, page 19 of 82. I suggest mentioning that results are presented by “mean \pm SE” at the methods section not in the results.
- 7) Line 10-11, page 19 of 82. Instead of 19.7 (\pm 0.2) m/s, I suggest presenting the values as 19.7 \pm 0.2.
- 8) Line 40, page 19 of 82. Please define what wing loading means the first time the concept is mentioned not here.
- 9) Figure 7, line 53 (Legend). The “l” is missing in slightly. Additionally, I suggest incorporating the 8 s and 12 s curves.
- 10) Line 45, page 24 of 82. I am not quite sure if this sentence should go between (). In case it does, there is not need to capitalize “Other” and there is an extra “.” Before the “(“ and before the “)“.

Discussion

- 11) Line 24, page 28 of 82. I am not sure if the sentence should go between ().
- 12) Line 18, page 29 of 82. 245 m/s ?

===PREPARING YOUR MANUSCRIPT===

one version should clearly identify all the changes that have been made (for instance, in coloured highlight, in bold text, or tracked changes);

===PREPARING YOUR REVISION IN SCHOLARONE===

To revise your manuscript, log into <https://mc.manuscriptcentral.com/rsos> and enter your Author Centre - this may be accessed by clicking on "Author" in the dark toolbar at the top of the

page (just below the journal name). You will find your manuscript listed under "Manuscripts with Decisions". Under "Actions", click on "Create a Revision".

-- If you are requesting an article processing charge waiver, you must select the relevant waiver option (if requesting a discretionary waiver, the form should have been uploaded, see 'File upload' above).

-- If you have uploaded any electronic supplementary (ESM) files, please ensure you follow the guidance at <https://royalsociety.org/journals/authors/author-guidelines/#supplementary-material> to include a suitable title and informative caption. An example of appropriate titling and captioning may be found at https://figshare.com/articles/Table_S2_from_Is_there_a_trade-off_between_peak_performance_and_performance_breadth_across_temperatures_for_aerobic_scope_in_teleost_fishes_/3843624.

At the 'Review & submit' step, you must view the PDF proof of the manuscript before you will be able to submit the revision. Note: if any parts of the electronic submission form have not been

completed, these will be noted by red message boxes - you will need to resolve these errors before you can submit the revision.

Author's Response to Decision Letter for (RSOS-211364.R3)

See Appendix D.

Decision letter (RSOS-211364.R4)

Dear Dr Richardson:

I am pleased to inform you that your manuscript entitled "Observations and Models of Across-wind Flight Speed of the Wandering Albatross" is now accepted for publication in Royal Society Open Science.

Please ensure that you send to the editorial office the individual files for each table included in your manuscript. You can send these in a zip folder if more convenient. Failure to provide these files may delay the processing of your proof.

Please remember to make any data sets or code libraries 'live' prior to publication, and update any links as needed when you receive a proof to check - for instance, from a private 'for review' URL to a publicly accessible 'for publication' URL. It is also good practice to add data sets, code and other digital materials to your reference list.

Royal Society Open Science is a fully open access journal. A payment may be due before your article is published. Our partner Copyright Clearance Center's RightsLink for Scientific Communications will contact the corresponding author about your open access options from the email domain @copyright.com (if you have any queries regarding fees, please see <https://royalsocietypublishing.org/rsos/charges> or contact authorfees@royalsociety.org).

on behalf of Dr Agustina Gómez-Laich (Associate Editor) and Professor Miles Padgett (Subject Editor).

Follow Royal Society Publishing on Twitter: @RSocPublishing
Follow Royal Society Publishing on Facebook:
<https://www.facebook.com/RoyalSocietyPublishing/>
Read Royal Society Publishing's blog:
<https://royalsociety.org/blog/blogsearchpage/?category=Publishing>

Appendix A

AWAS new letter to editor October 18, 2021

Dear Andrew Dunn,

Thank you for your comments on our paper RSOS-211364. We have followed your suggestions by splitting the previous results section into separate results and discussion sections. In particular, we created a new discussion section of nine paragraphs, which discusses 1) the minimum airspeed for dynamic soaring, and 2) how and why a wandering albatross limits airspeed and the aerodynamic force on its wings in fast winds to be considerably below values predicted by the dynamic soaring model. We think this new discussion section makes sense and is easier to navigate, especially for those expecting the usual structure of RSOS sections.

We tried to correctly answer the request for information concerning article processing fees (we chose category 2, option c), but we offer an explanation here since the issue seems complicated. The University of Glasgow has “prepay open access membership” with the Royal Society, which we think entitles Ewan Wakefield (relevant author) to a discount. The University should cover whatever other charges remain due to Ewan Wakefield’s funding. We assume that we will have to pay VAT, but are not clear exactly how this will be accomplished at this point. We hope this is sufficient information so that we can submit our revised paper and have it reviewed. We would be pleased to further clarify our statement concerning article processing fees if our paper is eventually accepted for publication.

Thank you for your help with our paper,

Philip Richardson

Appendix B

Reference: Manuscript ID RSOS-211364.R1

Letter to the Editors of RSOS,

The following are replies to the comments provided by the reviewers and editors of Manuscript ID RSOS-211364.R1. We thank the reviewers and editors for carefully reading our manuscript and providing helpful critical comments. We respond to the reviewers below and by modifying the manuscript.

From: Royal Society Open Science <onbehalf@manuscriptcentral.com>

Sent: Monday, February 21, 2022 9:48 AM

To: Philip L Richardson <prichardson@whoi.edu>

Cc: journal-submit@datadryad.org <journal-submit@datadryad.org>; Philip L Richardson <prichardson@whoi.edu>; Ewan.Wakefield@glasgow.ac.uk; ewan_wakefield@yahoo.co.uk <ewan_wakefield@yahoo.co.uk>; WAKEFIELD, EWAN <ewan.wakefield@durham.ac.uk>

On behalf of Dr Agustina Gómez-Laich (Associate Editor) and Miles Padgett (Subject Editor) <openscience@royalsociety.org>

Associate Editor Comments to Author (Dr Agustina Gómez-Laich):
Comments to the Author:

Dear authors,

The manuscript entitled “Observations and models of across-wind flight speed of wandering albatross” has now been seen by two referees, both of whom considered that substantial revisions are necessary. Both reviewers have concerns about the statistical analyses. Reviewer #1 states several weak points, which are principally related to the fact that the resolution that the wind and the GPS data have are not sufficient to compare model assumptions with dynamic soaring trajectories. Additionally, in the face of the results this reviewer suggests discussing more in detail alternatives to dynamic soaring. Reviewer #2 states that the presented data is not sufficient to support the claims of the paper and suggests testing the hypothesis with open data from a previous study. This reviewer also questions the assumption that dynamic soaring cycles are 10 s long and is concerned with the fact that authors modify the dynamic soaring cycle turn amplitude but keep the period constant.

Reply: Thank you for the helpful summary and comments. We respond below in detail to the reviewers’ comments.

Reviewer comments to Author:

Reviewer: 1

Comments to the Author(s)

This manuscript presents simulated flight trajectories of wandering albatrosses in relation to cross-winds, and the calculated speeds are compared with GPS-tracks of real albatrosses making flights around South Georgia. The main features of the loopy trajectories of dynamic soaring (DS) are turn angle and

cycle period. The main question is how well DS can explain the ground speed in low and high wind speeds, and whether other mechanisms of extracting energy than DS come into play. Alternative mechanisms are gust soaring, swell soaring and slope soaring (the latter was introduced by Wilson (1975; *Nature*; a reference that strangely is missing in the present paper). There is a trend in the data that airspeed increases with wind speed, although there is an enormous scatter in the data (Fig 6). The main outcome of the paper (Fig 7) suggests that DS explains a minority of the observations. In the light of this it is somewhat surprising that the authors cling to DS as main explanation for albatross flight.

Reply: We believe that DS is the main source of energy extracted by the birds from wind used for sustained soaring over the ocean. We infer that the birds also exploit updrafts over waves for soaring when the wind is weak and wind shear is not sufficient for DS by itself. Because we believe that updrafts provide only a minor part of energy for the birds we tracked for typical conditions in the Southern Ocean we have not discussed it in detail. An earlier paper by Richardson (2011) [20] compares soaring using wind shear and updrafts over waves and gives references for both soaring techniques including Wilson (1975). We added a two-paragraph summary of those findings in the present paper and the Wilson (1975) reference. We note that the structure and amplitude of updrafts over waves and wind-wave interactions are complicated, and it is difficult to assess how effectively birds use updrafts. We believe that we have demonstrated with our DS model that in principle DS could result in much faster airspeeds than we measured (as also demonstrated by the extremely fast DS radio-controlled gliders soaring in the lee of mountain ridges). We hypothesize that the birds fly slower than the model predicted speeds because the model predicts the fastest possible airspeed in dynamic soaring and that the birds fly slower than this in order to limit aerodynamic force on their wings and to make soaring over ocean waves easier. We believe that the observed airspeeds are achieved primarily by DS and that the birds do not need extra energy from updrafts for the usual conditions found in the Southern Ocean since more than enough energy is provided by DS. The scatter of observations about the trend line is probably due to several things as mentioned in the paper—variations of wing loading, variations of flight maneuvers (foraging) during the approximately one-hour intervals between GPS positions, and possible mismatches between the wind measurements and actual bird locations, which could cause errors in wind speed that appear as errors in airspeed. The scatter does not invalidate our conclusions that DS is the primary source of energy for sustained soaring by wandering albatrosses.

A main weak point of this analysis is that the various model results (varying wind speed, turn angles and cycle period) yield overall cross-country speed, which is compared with overall observed cross-country performance.

Reply: We are not sure why this is a weak point. We think our use of a realistic DS model is a sensible way to help analyze the data, especially if accurate high-resolution (10 Hz) tracking and wind data are not available.

Another weak point is that winds are not directly measured, but interpolated from weather data, which prevents the estimation of gust strength and the characteristics of the vertical wind gradient.

Reply: Of course it would be preferable to analyze albatross flight using contemporaneous, high resolution wind data. However, such data are, as far as we know, simply unavailable. Indeed, we are unaware of any technique that could currently be used directly to measure wind over the vast areas of the ocean (11 million km² in our study) traversed by free-ranging albatrosses. It is therefore necessary to use the wind measurements made by satellite scatterometers, which give reasonable estimates of wind shear, although not high temporal and spatial resolution values. Eventually, there will be accurate high-

resolution wind and trajectory measurement series with which a study of the fine-scale details of the soaring maneuvers can be made. To date we are not aware of such an excellent series nor have we seen relevant detailed analysis of higher resolution time series that shed light on the variations of turn angle and period of turns with respect to wind speed (or bird airspeed). We believe we can make an important contribution by further analyzing the time series data we have.

The GPS data are not of sufficient time resolution (one position every 0.5-2 hours, which is good for overall flight routes but not sufficient to test the trajectories of DS cycles) that would be required to actually compare model assumptions with actual dynamic soaring trajectories. This may explain the large discrepancies between model calculations and data, which suggest that DS soaring is rarely the explanation of observed flight speeds. I think alternatives to DS are not discussed in the face of the results.

Reply: We now discuss the alternatives to DS, but we do not believe that the alternatives are key to understanding the birds' speeds in typical conditions. We believe the "discrepancies between model calculations and data" do not invalidate DS as an explanation of our observed flight speeds and that the birds limit the fast predicted DS airspeeds in order to limit airspeeds and accelerations in the maneuvers. We conclude that it is only in low wind speeds that the birds necessarily exploit updrafts over waves for sustained soaring.

Specific comments

P3, L46 how was the L:D ratio of 21 obtained? Appears rather high unless in ground effect.

Reply: Colin Pennycuik (2008) [35] modeled and computed the glide polar of a wandering albatross (and other birds) and gives the representative best glide ratio of 21.2 at cruise speed of 16 m/s. The model is a bit complicated; his 2008 book should be consulted for details. As far as we know these values are reasonable ones for wandering albatrosses, although as mentioned in our paper there are large variations of wing loading and cruise speeds of the birds about the representative values given.

P14, L31 Wing loading is weight (mg) per unit wing area and therefore has unit N/m².

Reply: Thank you, changed units to N/m².

P16, L55 Unnecessary (tautologic) to say lift force, since lift is by definition a force. Same holds for drag.

Reply: Thank you; "force" is now omitted in these cases.

P21, L8 How is across-wind different from the conventional cross-wind?

Reply: There is no difference. A brief survey of the uses of "cross-wind" and "across-wind" in the literature suggests that both are used. We prefer across-wind because it is a similar construction to along-wind, which we also use.

P21, L13 Don't you mean ground speed here?

Reply: Yes, ground speed. But, across-wind components of airspeed and ground speed are equal. How about just "...average speed...?"

P21, L40 small airspeed should be low airspeed. Reply: Ok, thank you.

P22, L18/19 245 m/s ??

Reply: Fast speeds of DS radio-controlled gliders reach 245 m/s as measured by radar guns (see refs 21 and 28). These fast speeds are obtained by gliders flown in the lee of mountain ridges where wind shear can be very large on windy days. Our model, which predicts such fast glider speeds also predicts that fast speeds (~ 50 m/s) are possible using the characteristics of wandering albatrosses and our observed winds. However, mechanical gliders' wings are much stronger than the birds' wings. Thus, we believe the birds limit acceleration by limiting airspeeds and the turn angles flown.

P23, L28 How is it possible to reach airspeed of 50 m/s?

Reply: See our response to your previous comment. DS is amazingly powerful, and one almost has to see and hear the fast radio-controlled gliders to appreciate this. We believe that in principle it would be possible for a bird or glider to reach 50 m/s using DS in sufficient wind and waves as our DS model predicts and as gliders have been measured to do. Of course, a strong high-performance glider ($L/D \sim 30/1$) could fly much faster than an albatross in similar winds.

Reviewer: 2

Comments to the Author(s)

Observations and Models of Across-wind Flight Speed of the Wandering Albatross

Overall, I found that the data presented was not sufficient to support the claims from the paper; I also found that a very strong and non-obvious hypothesis was made (namely keeping the dynamic soaring cycle time constant while modifying all other parameters), with a large influence on the conclusions of the article.

I recommend testing the hypotheses with data from [10].

The main stated paper contributions is a list of the following hypotheses:

- 1- that albatrosses utilize updrafts from water waves in low winds to sustain flight
- 2- that in high winds, albatrosses do not seek to maximize their airspeed, as that would result in excessive g-loading.
- 3- albatrosses limit their airspeed to 20 m/s
- 4- the way albatrosses limit their speed is by means of smaller turn angles.

1- and 2- have already been stated elsewhere e.g. in the beautiful [How do albatrosses fly around the world without flapping their wings? Richardson PL, 2011].

Reply: We think the present GPS and wind data and the DS model simulations demonstrate these contributions better than the earlier paper. We studied these data and our DS model in order to analyze what the birds were doing and how they achieve the documented airspeeds.

Claim 3- is based on a very few data points, so I don't find the statement very strong. In particular, I'd like to point the authors to [SUSTAINED FAST TRAVEL BY A GRAY-HEADED ALBATROSS (THALASSARCHA CHRYSOSTOMA) RIDING AN ANTARCTIC STORM, Catry P et al., 2004] where an albatross was recorded at an average groundspeed of 127 km/h (35 m/s). I recognize that ground speed and airspeed are not the same quantity, and that the albatross species are different in the two papers. Still, it makes me doubtful that wandering albatrosses actually limit their airspeed to 20 m/s.

Reply: We are aware of the Catry et al, 2004 paper but we do not see its relevance here. As the reviewer rightly points out, ground speed is fundamentally different to airspeed. The former results from the addition of a bird's airspeed and the speed of the body of air in which a bird is moving. Catry et al. showed convincing that the high ground speed attained by the grey-headed albatross tracked in their study resulted from exceptionally fast tail winds – i.e., there was no need to invoke unusually high airspeeds to explain their observations. To explain detail, Catry et al, 2004 say the typical airspeed of small albatrosses is 8.9 ± 3.9 m/s. If we add the typical airspeed (8.9 m/s) to two times the standard deviation of airspeed ($2 \times 3.9 = 7.8$ m/s) as an estimate of the fast airspeed limit we get 16.7 m/s. This is somewhat slower than our 20 m/s limit for wandering albatrosses but reasonable considering the different cruise airspeeds. If we add to this the estimated tail wind speed (20.5 m/s) we get 37.2 m/s, close to the 35.3 m/s top speed mentioned by Catry et al. Note that this value is considered to be the ground speed along the sinusoidal trajectory not the speed of the straight line between two positions, as our value is. This value (37.2 m/s) implies that the fast ground speed (35.3 m/s) was largely determined by the fast tail wind (20.5 m/s). Note that this estimated tail wind speed is very close to the maximum wind speed 20 m/s measured during our tracking. The fastest ground speed of our tracked wandering albatrosses is 22.3 m/s, which is in a downwind direction and includes a component of tail wind speed of ~ 7.2 m/s. Our conclusion is that the values mentioned by Catry et al are consistent with ours and that our value of wandering albatross airspeed limit of 20 m/s is reasonable.

Reply continued: It is clear that the fastest observed ground speeds of albatrosses are an order of magnitude lower than those predicted by dynamic soaring theory. That this theory is correct, has been proven experimentally using manmade gliders of similar sizes and dimensions to albatrosses which have attained airspeeds reaching 245 m/s in dynamic soaring flight (Richardson PL. 2012 High-speed dynamic soaring. Radio-Cont. Soaring Dig. 29, 36-49). The most parsimonious explanation for this discrepancy is that albatrosses do indeed limit their airspeeds, probably to avoid mechanic stress.

The nonlinear DS model is based on the strong assumption that dynamic soaring cycles would be 10s long. However, there is no strong argument for keeping that value fixed. The only justification I found in the paper is "typical observed [half-period] of 5 s", and a reference to a " ~ 11 s cycle in [15]". While it is true that [15] has mean cycles of 11-13s, the standard deviation on cycle duration is rather large.

Reply: We believe there are three different reasonable values of typical observed periods of the series of two connected turns in the typical DS maneuver of across-wind wandering albatross flight in the literature—10.7 s (Idrac, 1924 [24]); ~ 10 s (Sachs, 2016 [8]); and ~ 11 s (Fig. 7, Bousquet et al. 2018 [15]). In our paper we used a constant period of turn angle of ~ 5 s as being representative of half the ~ 10 -11 s typical maneuver period. Clearly this is an assumed average value. In support of the $t = 5$ s

assumption, there appears to be a rather small variation of turn angle vs ground velocity in the 1 Hz GPS trajectory [10] analyzed by Gabriel Bousquet as discussed below.

The authors make the hypothesis that with increasing airspeed the turn angle is reduced (with the implicit assumption that cycle time be kept constant). However, the data they use does not provide clear evidence of it. This said, I believe that the open data from [10] would allow actually testing the authors' hypothesis (hypothesis: cycle period is independent of turn amplitude -- I personally believe that the hypothesis does not hold and would instead bet that there is a negative correlation between turn amplitude and cycle time.)

Reply: Good point. As mentioned above we have tried to verify this hypothesis but have been unable to do so. Clearly it remains a hypothesis that needs to be verified when accurate high-frequency tracking and wind data are available and a serious detailed study of these data can be made. Indeed, surely one of the purposes of scientific papers (and indeed of the scientific method) is to confront theory with data and in so doing, generate new hypotheses.

This is an important point because fixing the cycle time to a constant value has a tremendous influence on all the conclusions of the paper. In particular, if one chose $t \sim TA$ in equation (B3), one would see that W is smaller for smaller values of TA .

Reply: Good points. Although there could be a trend of the period of turn angle as airspeed and wind speed increase, and similarly for turn angles vs wind speed, we do not know of data or results of studies that document these trends. We reached out to Gabriel Bousquet [15], who analyzed the Yonehara et al. 1 Hz GPS data [10], and asked him if he would please help us check to see if there were statistically significant relationships between the turn angle (TA) and wind speed (W) and also the turn period (t) and W for wandering albatrosses soaring in the across-wind direction. His paper [15] included a histogram of TA 's (our figure 3) and gave the median and average t but did not establish their correlations with respect to W or airspeed.) Recently, Bousquet generously provided us with his analysis of the 1Hz GPS tracking data, which derived time series of t , TA , and U along trajectories for the case of a wandering albatross soaring across-wind. At present the derived wind speeds are not available, but Bousquet tells us he will try to find time to calculate them.

Reply continued below in several paragraphs: We made plots of the 1 Hz derived data and discovered some apparent problems that make a detailed analysis and comparison with our paper's assumptions and conclusions problematic. In particular, the derived data are at a frequency of 1 Hz, which is not sufficiently large to well-resolve small t 's and associated TA 's. The t (and also TA and U) values are cut off for t below 2.5 s. We infer there could be significant numbers of values located below 2.5 s. The omission of these values could influence mean and median values and the sign and magnitude of the correlations. In addition, there is a very large variability of values of t , TA , and U with some very large values of TA including those over 180 deg, which is the maximum possible for dynamic soaring (DS) and very much larger than the median and average values. And there are some large values of U up to ~ 50 m/s (2.5 times larger than our fastest across-wind airspeed of 20 m/s). The large values and large variability raise questions about GPS measurement errors and resulting errors in the time series. We have not yet been able to sort out these issues to our satisfaction. A major issue is how to identify and omit outliers that could be due to measurement errors. We conclude that the derived 1 Hz data cannot be used to prove or disprove our assumption of constant t of 5 s and our inference of decreasing TA as W increases. There are possibly other higher resolution data sets that could be analyzed for more information, but finding and obtaining the data and doing detailed analyses is not possible in the

relatively short amount of time before the revised manuscript is to be submitted. Therefore, we believe the derived data cannot (at this time) be used to establish accurate trends of period of turns vs wind speed and bird speed, and similarly for turn angles.

Below we describe in some detail our preliminary analysis of the 1 Hz data provided by Bousquet. Our analysis uses two different series: The first series includes all of the 10,622 original derived values of t , TA and U, including values of U ranging from around 5-50 m/s. The second series is a subset consisting of 6,530 values of t , TA, and U, within the range of U from 10-20 m/s. This subset includes the main cluster of U values and excludes large values ($U > 20$ m/s). The 10-20 m/s range of U values is roughly equivalent to the range of most of the across-wind airspeeds reported in our paper.

Linear least square best fits to t vs U and TA vs U for (1) all data and (2) a subset of data are given below:

All data: $t = 8.4 \pm 0.2 - (0.08 \pm 0.01) * U$, where t is in seconds, U is in m/s, and 95% CI.

All data: $TA = 65 \pm 8 + (1.1 \pm 0.4) * U$, where TA is in degrees.

Subset of data: $t = 9.0 \pm 0.6 - (0.12 \pm 0.04) * U$.

Subset of data: $TA = 141 \pm 14 - (3.8 \pm 0.9) * U$.

Our interpretation of the linear model of t vs U using all the 1 Hz data (above) indicates a relatively small decrease of t over the 10 m/s range of most values of U (between 10 and 20 m/s). The median t is 5.8 s (quite close to our assumed value of 5.0 s) and the decrease in t over 10 m/s is $\sim 0.8 \pm 0.1$ s (95% CI). This result could be considered somewhat in agreement with our assumed constant 5 s. A linear model indicates TA increases by around 11 ± 4 deg over a 10 m/s increase of U, (compared to median TA of 67 deg) which contradicts our hypothesized decrease of TA vs U. The increase of TA appears to mainly be due to TA values corresponding to $U > 20$ m/s.

A linear model of t vs U for the data subset (between $U = 10-20$ m/s) indicates that t decreases around 1.2 ± 0.4 s over 10 m/s compared to the median of t of 6.2 s. The median value of t (6.2 s) using the subset of data is somewhat larger than t (5.8 s) using all the data, but still the variation of t is relatively small (over 10 m/s) roughly in agreement with our assumed constant $t = 5$ s. The TA was found to decrease around $38 \text{ deg} \pm 9 \text{ deg}$ over 10 m/s compared to the median of 68 deg. This result is also in rough agreement with our conclusion about a decrease of TA as wind speed increases in order to limit acceleration. The relatively large decrease of TA (normalized by median TA) = $-56\% \pm 13\%$ tends to counter the increase of U (normalized by the median U) of $+63\%$ over 10 m/s. This suggests that a decrease in centripetal acceleration (-56%) due to the decrease of TA nearly balances the increase of centripetal acceleration ($+63\%$) due to the 10 m/s increase of U (over 10-20 m/s). (Centripetal acceleration in a steady balanced turn is proportional to $U * TA / t$). This result is in general agreement with our paper about a bird decreasing TA to limit acceleration as its velocity increases. We note, however, that these values are based on a subset of the 1 Hz data and disagree with results using all the data. This raises questions about how much significance we should place on these results.

We know that one should be very careful about choosing which outlier values to exclude when analyzing data. We caution that the apparent rough agreement of the 1 Hz TA data with our paper's conclusions appears to be partially a result of our choice of the 10-20 m/s U limits for the analysis (which excludes $U > 20$ m/s). Other choices of which data to include could lead to different results. It is difficult to place much significance on our results without further understanding how GPS measurement errors could lead to errors in the time series of t , TA, and U. We are still seeking a good justification of which data to exclude.

Reviewer 2 comments continued: In figure 6 (the main figure of article), there is a lot of scatter. I'm curious whether the scatter is in part due to the breadth of flight directions "45 to 135". It would be nice to color-code the scatter point with flight direction. Is there a pattern?

Reply: Good question and suggestion. We tried several plots using colors to represent different relative flight directions. We were not able to find any obvious relevant patterns. We think that using airspeed data for the 45-135° relative angles (with respect to the wind direction) provides a fairly large data set that is representative of the average across-wind airspeed.

Journal Name: Royal Society Open Science

Journal Code: RSOS

Online ISSN: 2054-5703

Journal Admin Email: openscience@royalsociety.org

Journal Editor: Andrew Dunn

Journal Editor Email: openscience@royalsociety.org

MS Reference Number: RSOS-211364.R1

Article Status: SUBMITTED

MS Dryad ID: RSOS-211364.R1

MS Title: Observations and Models of Across-wind Flight Speed of the Wandering Albatross

MS Authors: Richardson, Philip; Wakefield, Ewan

Contact Author: Philip Richardson

Contact Author Email: prichardson@whoi.edu

Contact Author Address 1: 360 Woods Hole Road

Contact Author Address 2:

Contact Author Address 3:

Contact Author City: Woods Hole

Contact Author State: Massachusetts

Contact Author Country: United States

Contact Author ZIP/Postal Code: 02543-1050

Keywords: wandering albatross, GPS tracking, dynamic soaring, wind shear, airspeed, flight trajectory

Abstract: Wandering albatrosses exploit wind shear by dynamic soaring, enabling rapid, efficient, long-range flight. To explore this flight mode, we compared the ability of a nonlinear dynamic soaring model and a linear empirical model to explain observed variation of the airspeeds of GPS-tracked albatrosses in across-wind flight. In fast winds (> 8 m/s), maximum observed airspeeds reach an asymptote at ~ 20 m/s, whereas the dynamic soaring model predicts much faster airspeeds, up to around 50 m/s. We hypothesize that the birds actively limit airspeed by making fine-scale adjustments to turn angles and soaring heights. Predicted dynamic soaring airspeeds do not extend down to the slowest winds (< 3.2 m/s) of observed flight. We hypothesize that in slow winds wandering albatrosses obtain additional energy from updrafts over water waves. The dynamic soaring model predicts that the minimum wind speed necessary to support dynamic soaring at a cruise airspeed of 16 m/s is 3.2 m/s, achieved via a flight trajectory of linked 137° turns. In reality, observed turn angles are typically ~ 60°. Our simulations suggest that birds may necessarily use smaller turns angles than the theoretical optimum for fast flight in order to limit aerodynamic force on their wings.

EndDryadContent

Replies to reviewers of MS ID RSOS-211364.R2 October 3, 2022

We thank the reviewers and Associate Editors and for carefully reading the revised manuscript and offering helpful comments to improve it. Below we reply to the comments and hope we have sufficiently improved the paper by additions and modifications based on the reviews.

Reviews and replies:

Associate Editor Comments to Author (Dr Agustina Gómez-Laich):

Associate Editor: 1

Comments to the Author:

Dear authors,

Thank you for this new version of the Ms entitled “Observations and models of across-wind flight speed of wandering albatross”. I appreciate all the efforts you made answering the questions and concerns raised by the reviewers. The paper reads much more clearly now, however, the Ms it is not acceptable for publication in the present form.

The three reviewers raised concerns about the simulation and analysis, particularly regarding the dissimilarity of the spatial and time scale employed for the models and for the validation. Additionally, Reviewer#1 as well as Reviewer#3 raised concerns about the 10s-cycle hypothesis. Regarding this particular issue, Reviewer#3 suggests to present models where $t = 4.5, 5, \text{ and } 5.5$, to illustrate the effect that varying cycle duration would have on the model predictions. Reviewer#1 also has doubts about the numerical value "20 m/s" for the upper airspeed. Reviewer#3 also suggests expanding at least a bit on the effect that varying bank angles might have in context of the presented model.

Kind regards,
Agustina

Reply: Thank you for the summary of comments. We address these issues below.

Associate Editor: 2

Comments to the Author:

Dear authors,

I appreciate all the efforts you made answering the questions and concerns raised by the reviewers. The paper reads much more clearly now, however, since the corrections involved even preliminary analyses with a new data set, I suggested sending the Ms. back to review.

Reviewer comments to Author

Reviewer: 2

Comments to the Author(s)

Comment:

Thank you for the thoughtful answer. I appreciate your further data analysis. In particular I appreciate that you more explicitly discuss that it is a hypothesis that demands more study.

Comment:

While there are points where I still do not align with the authors, I am more comfortable if the article given the modifications, and author's responses message.

Comment:* There is still a spot in the article where the 10s-cycle hypothesis makes me uncomfortable with your analysis: the paragraph on "p25 of 86", line 14 ("In general, the DS model predicts that smallest values of..."). If you were not constraining the cycle duration to 10s, you would obtain a different conclusion (for instance, I bet that if you tried the exact same model with a smaller cycle type, for instance 8s, you'd get a smaller W_{min} , wherein the optimal cycle would be completed at smaller turn angle.

Reply: Thank you for the suggestion. We have clarified the paper by now using a DS cycle period of two turns, which we now call t in the revised paper. We hope that this change is more consistent with previous usage. We recalculated W_{min} using a cycle period $t = 8$ s, and as you suggested we find that the W_{min} is indeed smaller (2.9 m/s) than that for $t = 10$ s (3.2 m/s) and the turn angle is also smaller (125° vs. 137°). We find that the across-wind speed is larger (13.0 m/s) than for 10 s (12.4 m/s) and that lift (1.4 g) is slightly larger than for $t = 10$ s (1.3 g) but should be manageable for a bird in low winds. Therefore, we consider it a reasonable hypothesis for a bird to start dynamic soaring in wind speed of around 2.9 m/s with a turn angle of 125° and $t = 8$ s. The median $t = 11.5$ s given by Bousquet et al. is close to 12 s. This value is representative of the across-wind section of trajectory 4 which Bousquet et al. estimate as having a mean velocity of around 20 m/s (950 kilometers/9 hours). We note that this speed agrees with the estimated maximum across-wind airspeed ~ 20 m/s of our tracked albatrosses. We now mentioned this information in the paper.

Comment:* I also still have doubts on the paper, in the numerical value "20 m/s" for the upper airspeed of the albatross. I appreciate The author's reply regarding Catry and al., where the author ran a computation to prove the their "20 m/s" claim was compatible with the observations from Catry et al. Indeed the author does a pure algebraic addition, while the airspeed to consider should rather be $\sqrt{35^2 - 20.5^2} = 28$ m/s.

Reply: We interpret the Catry et al. paper's saying that the bird "experienced consistent tail winds with an estimated speed of 70-80 km/hour" to mean that the bird was flying in a downwind direction and not across-wind. For a bird flying in a downwind direction, airspeed =

ground speed – tail wind speed. This would result in airspeed = $35 \text{ m/s} - 20.5 \text{ m/s} = 14.5 \text{ m/s}$ (using the reviewer's values of speed). This airspeed is below our maximum 20 m/s, so there does not appear to be a conflict with our conclusion about maximum airspeed. We should note that our maximum airspeed of 20 m/s is for across-wind flight not downwind flight, which appears to be the case for the bird mentioned by Catry et al.

Reply continued: The reviewer's calculation assumed that the air velocity of the albatross was perpendicular to the wind velocity, which resulted in an estimated airspeed of 28 m/s. This implies that the direction of ground velocity would be $\sim 54^\circ$ degrees relative to the downwind direction, or closer to an across-wind direction than a down-wind direction. The problem of sorting out whether the estimated airspeed is larger or smaller than 20 m/s requires better information about the value of the direction of the ground velocity relative to the downwind direction, information we do not have. With the information available one cannot determine whether this bird's estimated airspeed would be smaller or larger than 20 m/s.

Comment; I fully agree with the author that "It is clear that the fastest observed ground speeds of albatrosses are an order of magnitude lower than those predicted by dynamic soaring theory"; I predict though that the "20 m/s" value is too low. I suspect that Yonehara's data could be analyzed further to show this -- but I understand if this is "left for future work".

Reply: The average ground velocity of the fast across-wind section of the Yonehara et al. trajectory 4 was estimated by Bousquet et al. as being $\sim 20 \text{ m/s}$ (as mentioned above), which agrees with our estimated maximum airspeed value of 20 m/s. We clearly leave for future work more analysis of the Yonehara et al. data. Our preliminary analysis indicates that the data have large variability in ground velocity, cycle period and turn angle despite the average velocity of around 20 m/s. The detailed trajectories look as if GPS errors are a problem in resolving the individual turns and this possibly creates part of the large variations. A detailed analysis of the data along the lines hypothesized here is problematic. We added some of this to the paper.

Comment: Accordingly I would be delighted if it was made more explicit in the article that 20 m/s is not an absolute upper bound, but the maximum seen in that particular dataset (particularly in the abstract).

Reply: Good point. Added to paper.

Reviewer: 1

Comments to the Author(s)

The authors have prepared a revision of the submitted manuscript. I have carefully read the authors replies to both my and the other reviewers comments, and it is clear from this that the authors have been unable to produce a publishable manuscript. The reply contains multiple "we believe.." explanations to many of the comments, which I do not consider sufficient. It is clear that the uncertainties regarding the simulations and suggested further analyses that the authors have not made for the revision, and that in combination with the mismatch between sampling rates and estimated quantities, suggest to me that publication should be deferred until a more

satisfying analyses can be presented. The argument that appropriate wind data are not available at the necessary time and spatial resolution, does not implicate that one should go on as if the data available are useful anyway.

Reply: The purpose of our analysis is firstly to test the adequacy of a simple theoretical model of dynamic soaring to predict the mean across-wind airspeed of wandering albatrosses and secondly, as per the scientific method, to refine and revise hypotheses in the light of our findings. While relatively low temporal resolution, the tracking data we use to do this is adequate to show that the model overestimates airspeeds across a range of wind speeds and predicts no dynamic soaring at low wind speeds when the tracked birds were observed in flight. In the revised manuscript, we explore whether this is due to a failing in our assumptions about turn angles and the period of the dynamic soaring cycle. Naturally, this leads to further hypothesized combinations of parameters that would lead to the model better conforming to the data. We view these hypotheses as one of the main outcomes of our analysis and trust this will stimulate others to test them as higher resolution biologging and wind measurement techniques become available.

Reviewer: 3

Comments to the Author(s)

Thank you for the opportunity to review this very interesting and well written manuscript. The proposed model of across-wind flight speed of dynamic soaring albatrosses will be useful in shaping our understanding of this incredible flight mode.

Although I agree with most of the conclusions in this manuscript, I do not believe the concerns raised by the first two reviewers were adequately addressed. I do believe that albatrosses limit their airspeed to reduce the physical force on their wings, but I don't believe that the validation data (airspeed derived from coarse-scale GPS loggers) are adequate to test these predictions. The model predicts airspeed of a single DS cycles (i.e. seconds) whereas the validation data are on a much larger scale (minutes to hours). The analysis of 1 Hz data is presented in the rebuttal, but I could not see how this was incorporated in the manuscript. I believe that these data have to be presented to the reader to make clear that the GPS data we currently have is not necessarily adequate to test these hypotheses.

Reply: Please see the reply to Reviewer 1 above. Our preliminary analysis of the Yonehara et al. 1 Hz GPS data found they were inadequate to validate our hypotheses, and therefore we have not included the data in the paper. We are not aware of any available data with sufficient quantity, accuracy and resolution to validate our hypotheses, and we have not seen results of studies that would be helpful in this regard. We hope that these kinds of data will eventually be obtained and used to prove or disprove our hypotheses and to further reveal what albatrosses do in their complicated dynamic soaring maneuvers over the real ocean. We mention some of this in the paper.

Comment: Secondly, the fixing of DS duration at 10 s was also raised by both reviewers. A possible suggestion would be to present models where $t = 4.5, 5, \text{ and } 5.5$, to illustrate the effect

that varying cycle duration would have on the model predictions, e.g. Fig 7 with multiple model prediction lines.

Reply: Thank you for the suggestion. We now evaluate the assumption of different cycle periods (8, 10, and 12 s) with our dynamic soaring model. As mentioned above we now estimate the effect of an 8 s cycle period on the minimum wind speeds required for initiating dynamic soaring. And we evaluate how increasing cycle period (and decreasing turn angle) can contribute to reducing the aerodynamic acceleration of lift in a steady horizontal turn. We note that varying cycle period does not directly cause variations of across-wind airspeed, although decreasing turn angle does increase across-wind airspeed, which would seem advantageous for an albatross foraging over the ocean because a bird could search a wider area in a given amount of time. Therefore, increasing cycle period (as wind speed increases) would have minor influence on the slope of the line of airspeed vs. wind speed that includes decreasing turn angles (Fig. 8). This information is now mentioned in the text.

Reply continued: We modeled the effect of varying the cycle period and considered adding prediction lines to Fig. 7 for 8 s and 12 s cycle periods. However, we think the results can be summarized by noting that the three similar predicted curves for 8, 10, and 12 s begin with an across-wind airspeed of 14.4 m/s, that the 8 s curve is shifted slightly to the left of the 10 s curve from $W = 3.7$ m/s to $W = 3.2$ m/s, and the 12 s curve is shifted slightly to the right of the 10 s curve from $W = 3.7$ m/s to $W = 4.3$ m/s (now mentioned in caption to fig 7). We think that the addition of these lines to Fig. 7 would tend to add clutter and not be very informative.

Comment: Lastly, it might be worth expanding a bit on the effect that varying bank angles might have in context of this model. From what I could gather, the model assumes a constant banked turn. Recent studies have been able to measure fine-scale bank angle of free ranging dynamic soaring birds (see refs below). It would be interesting to read your thoughts on how future studies could look at bank angles and how these would influence the turn angles of a DS cycle at varying wind speeds.

Reply: Good question. Accurate and high temporal resolution (10 Hz) 3-d GPS measurements, like those described by Sachs et al. (2016), accompanied by accurate accelerometers, videos of bank angles, and good measurements of local winds and waves could be a big help in helping us learn about the detailed 3-d flight maneuvers of wandering albatrosses including bank angle, turn angle, cycle period. These kinds of measurements are difficult to obtain in sufficient numbers to be useful. Analyses of the different data sets will require a lot of thought and hard work. I look forward to seeing new results over the next ten years or so. Clearly, our dynamic soaring model is a simplification of a bird's real trajectory over the ocean. More complicated models will need to include new information about a bird's trajectory and the local wind and waves. These more advanced models could be extremely useful in revealing the detailed dynamics of flight including airspeed, lift, bank angle, and turn angle as a function of wind speed.

Reply continued: The Yonehara et al. 1 s GPS data that we examined give a flawed measure (due to GPS errors and gaps) of the variations of ground speed, turn angle and turn duration on the local scale (seconds-minutes). The large local variations in ground speed can be inferred to be due to a bird climbing and descending across the wind-shear layer and turning at the top and

bottom of the climb. Yonehara et al. used variations of ground speed interpreted as leeway to infer wind speed, but trading altitude (PE) for speed (KE) in a DS maneuver could also be relevant. Unfortunately, the 1 Hz GPS measurements are limited to horizontal trajectories since the data are not accurate enough to resolve variations in height. All the measurements of ground speed, turn angle, and cycle period have large variability that can be interpreted in part to a bird responding in complicated ways to local wind and wave fields. Analysis can be difficult due to the large variability and inferred GPS errors and gaps. Clearly more accurate and higher resolution GPS measurements would be very helpful in interpreting trajectories. Some of the above was added to the text.

References:

S. Schoombie, J. Schoombie, C. W. Brink, K. L. Stevens, C. W. Jones, M. M. Risi, P. G. Ryan, Automated extraction of bank angles from bird-borne video footage using open-source software. *J. Field Ornithol.* 90, 361–372 (2019).

Kempton, James A., et al. "Optimization of dynamic soaring in a flap-gliding seabird affects its large-scale distribution at sea." *Science Advances* 8.22 (2022): eabo0200.

Reply: Thank you for the references, which we now include in the paper

Appendix D

Reply to latest comments from the Editors and Reviewers concerning Manuscript RSOS-211364.R3

Dear Editors and Reviewers,

Thank you for your additional comments and suggestions. We reply in bold text below to each of your comments.

Dear Dr Richardson

On behalf of the Editors, we are pleased to inform you that your Manuscript RSOS-211364.R3 "Observations and Models of Across-wind Flight Speed of the Wandering Albatross" has been accepted for publication in Royal Society Open Science subject to minor revision in accordance with the referees' reports. Please find the referees' comments along with any feedback from the Editors below my signature.

Please submit your revised manuscript and required files (see below) no later than 7 days from today's (ie 25-Oct-2022) date. Note: the ScholarOne system will 'lock' if submission of the revision is attempted 7 or more days after the deadline. If you do not think you will be able to meet this deadline please contact the editorial office immediately.

on behalf of Dr Agustina Gómez-Laich (Associate Editor) and Miles Padgett (Subject Editor)
openscience@royalsociety.org

Associate Editor Comments to Author (Dr Agustina Gómez-Laich):

Associate Editor

Comments to the Author:

Dear authors,

Thank you for this new version of the Ms. I appreciate all the efforts you made answering the questions and concerns raised by the reviewers. I only have a few minor comments.

Introduction.

1) Even though this paper does not follow the classic structure of a manuscript, I suggest authors to refresh the readers at the end of the introduction which are the principal aims of the work.

Reply: We added two sentences at the end of the introduction about the principal aims of the work: “The purpose of our study is to test the adequacy of a simple theoretical model of dynamic soaring, to predict the mean across-wind airspeed of wandering albatrosses, and to revise hypotheses in the light of our findings. Our aims are firstly, to investigate the relationship between mean velocity of wandering albatrosses and wind velocity in order to gain a better understanding of their long-range flight characteristics; secondly, to use model simulations to interpret flight observations in terms of the period and turn angle of a dynamic soaring cycle; and thirdly, to explain how and why wandering albatrosses fly so much slower than the fast speeds achieved by dynamic soaring albatross-sized gliders.”

2) Line 54-55, page 12 of 82 (tracked changes version). A space is missing between “to” and “16 m/s”

Reply: Done, thank you.

3) Line 54-55, page 13 of 82 (tracked changes version). “In” should be replaced by “is”. That is to say “variability of turn angles is indicated instead of “in indicated”.

Reply: Done

Methods.

4) Line 41, page 17 of 82 (tracked changes version). Hence we use “the” term. “The” is missin

Reply: Done

Results.

5) Line 5, page 19 of 82. Please eliminate the “)” that is before [11]

Reply: Done

6) Line 8-9, page 19 of 82. I suggest mentioning that results are presented by “mean±SE” at the methods section not in the results.

Reply: Done

7) Line 10-11, page 19 of 82. Instead of 19.7 (± 0.2) m/s, I suggest presenting the values as 19.7± 0.2.

Reply: Done

8) Line 40, page 19 of 82. Please define what wing loading means the first time the concept is mentioned not here.

Reply: The first time “wing loading” is mentioned is at this location, and so we kept the definition of it here.

9) Figure 7, line 53 (Legend). The “l” is missing in slightly. Additionally, I suggest incorporating the 8 s and 12 s curves.

Reply: “l” was added. We considered incorporating the two other curves earlier and decided it is simpler and less cluttered to omit them. We just recently reconsidered adding them, but there is a problem in that the graphics designer who created the figure is very busy with more important work for the administration (she says) and there would be a considerable wait until the changes could be accomplished. We were given only one week to make these late changes. Therefore, we have abandoned trying to change the figure but could proceed if it is deemed to be very important.

10) Line 45, page 24 of 82. I am not quite sure if this sentence should go between (). In case it does, there is not need to capitalize “Other” and there is an extra “.” Before the “(“ and before the “)”.

Reply: Fixed

Discussion

11) Line 24, page 28 of 82. I am not sure if the sentence should go between ().

Reply: Fixed

12) Line 18, page 29 of 82. 245 m/s ?

Reply: We do not understand why the question mark was added after 245 m/s. We believe that we need to point out how fast the albatross-sized RC gliders can fly using dynamic soaring, and so we give the top recorded ground speed of 245 m/s. This was measured with a radar gun like those used by police to measure speeding cars. We give a reference to a

much more detailed explanation of fast RC gliders in [22] and the listing of numerous other fast RC gliders and more details in [28] in case readers would like to explore this subject. We did not think it appropriate to go into much detail in the text because wandering albatrosses do not use dynamic soaring to go very fast compared to these gliders. We think our mention in the text is sufficient to document the fast RC glider flight without going to a lot of detail.